# HoT-VI: Reparameterizable Variational Inference for Capturing Instance-Level High-Order Correlations

**Junxi Xiao**[1]    **Qinliang Su**[1,2*]    **Zexin Yuan**[1]

[1]School of Computer Science and Engineering, Sun Yat-sen University, Guangzhou, China
[2]Guangdong Key Laboratory of Big Data Analysis and Processing, Guangzhou, China
{xiaojx7,yuanzx7}@mail2.sysu.edu.cn suqliang@mail.sysu.edu.cn

## Abstract

Mean-field variational inference (VI), despite its scalability, is limited by the independence assumption, making it unsuitable for scenarios with correlated data instances. Existing structured VI methods either focus on correlations among latent dimensions which lack scalability for modeling instance-level correlations, or are restricted to simple first-order dependencies, limiting their expressiveness. In this paper, we propose **H**igh-**o**rder **T**ree-structured **V**ariational **I**nference (HoT-VI)[2], that explicitly models $k$-order instance-level correlations among latent variables. By expressing the global posterior through overlapping $k$-dimensional local marginals, our method enables efficient parameterized sampling via a sequential procedure. To ensure the validity of these marginals, we introduce a conditional correlation parameterization method that guarantees positive definiteness of their correlation matrices. We further extend our method with a tree-structured backbone to capture more flexible dependency patterns. Extensive experiments on time-series and graph-structured datasets demonstrate that modeling higher-order correlations leads to significantly improved posterior approximations and better performance across various downstream tasks.

## 1 Introduction

Variational inference (VI) is a widely used framework for approximating the posterior distribution in latent-variable models $p_{\boldsymbol{\theta}}(\mathbf{X}, \mathbf{Z}) = p_{\boldsymbol{\theta}}(\mathbf{X}|\mathbf{Z})p(\mathbf{Z})$, where $\mathbf{X} = [\mathbf{x}_1, \mathbf{x}_2, \cdots, \mathbf{x}_N]$ and $\mathbf{Z} = [\mathbf{z}_1, \mathbf{z}_2, \cdots, \mathbf{z}_N]$ are observed data and the corresponding latent variables, respectively. VI seeks to approximate the intractable posterior $p(\mathbf{Z}|\mathbf{X})$ with a tractable surrogate distribution $q_{\boldsymbol{\phi}}(\mathbf{Z}|\mathbf{X})$ from a parametrized distribution family $\mathcal{Q}_{\boldsymbol{\phi}}$, by maximizing the evidence lower bound $\mathcal{L}(\boldsymbol{\theta}, \boldsymbol{\phi}) = \mathbb{E}_{\mathbf{Z} \sim q_{\boldsymbol{\phi}}}[\log p_{\boldsymbol{\theta}}(\mathbf{X}, \mathbf{Z}) - \log q_{\boldsymbol{\phi}}(\mathbf{Z}|\mathbf{X})]$. In typical settings where data instances are assumed to be independent, the joint distribution naturally factorizes across instances as $p_{\boldsymbol{\theta}}(\mathbf{X}, \mathbf{Z}) = \prod_i p_{\boldsymbol{\theta}}(\mathbf{x}_i, \mathbf{z}_i)$ where $\mathbf{x}_i$ and $\mathbf{z}_i$ denote the $i$-th data instance and latent variable. Under these scenarios, we can reasonably use the mean-field posterior $q_{\boldsymbol{\phi}}(\mathbf{Z}|\mathbf{X}) = \prod_i q_{\boldsymbol{\phi}}(\mathbf{z}_i|\mathbf{x}_i)$ for model inference and training. However, many real-world datasets exhibit complex relationships among instances, making the independence assumption across data instances untenable. For instance, in multivariate time series [59, 21], the latent states $\mathbf{z}_i$ at a timestamp may depend on those preceding and succeeding timestamps. Similarly in graph-structured data such as social networks [19] and citation graphs [29], the latent representations $\mathbf{z}_i$ and $\mathbf{z}_j$ of connected nodes are typically correlated due to underlying relational structure. In these scenarios, the mean-field posterior is clearly inadequate, as it ignores crucial dependencies among data points.

---

*Corresponding author.
[2]Code is available at: https://github.com/Mephestopheles/HoT-VI.

39th Conference on Neural Information Processing Systems (NeurIPS 2025).

Many existing variational inference methods have attempted to incorporate structured dependencies into posterior approximation [68, 3, 31], but most of these approaches focus on modeling correlations among dimensions within a latent variable. Since the number of latent dimensions is typically small (*e.g.*, dozens to hundreds) compared to the size of datasets (*e.g.*, thousands to millions), methods targeting dimension-level correlations cannot be applied to capture instance-level correlations, especially for large datasets. Some recent efforts have been devoted to explicitly model instance-level correlations. For example, Correlated Variational Autoencoder (CVAE) [56] introduces a tree-structured variational distribution that captures pairwise dependencies between neighboring latent variables. Similarly, Tree-structured Variational Inference (TreeVI) [63] builds a correlation matrix derived from a tree structure over instances, enabling efficient sampling and scalable inference. However, the reliance on capturing pairwise interactions limits these methods to first-order correlations, and precludes the representation of cyclic or higher-order dependency structure among latent variables. Yet in many real-world domains such as financial time series [51], sensor networks [12], climate data [40] and evolving graphs [24], correlations among data instances are not merely pairwise. These data frequently exhibit high-order dependencies, where the relationship between two latent variables is mediated by the joint influence of multiple others. In time series, for instance, the latent state at a given time point may not only depend on nearby time steps, but also on patterns that occurred further in the past [17]. In such cases, methods constrained to pairwise or tree-structured dependencies are fundamentally limited in expressiveness, necessitating a more expressive variational inference framework capable of capturing higher-order instance-level dependencies.

In this work, we introduce a novel variational inference framework that overcomes the limitations of existing methods by incorporating higher-order dependency structures among latent variables. Rather than restricting attention to first-order correlations, our approach is able to capture more expressive $k$-order dependencies. We theoretically show that, by imposing a $k$-order dependency structure into the global variational posterior, the high-dimensional global posterior can be expressed in terms of a set of $k$-dimensional local marginal distributions. By leveraging these local marginals, a sequential sampling method is developed to draw parameterized samples from the high-dimensional global variational posterior, which are then substituted into the evidence lower bound for training. To ensure the validity of this approach, conditional correlations are introduced to re-parameterize the correlation matrices of local marginals. We prove that by using the conditional correlations to represent the correlation matrices, the positive definiteness of correlation matrices are guaranteed, and so does the validity of the developed VI approach. Later, we further show that the approach can be extended to support more structured dependency patterns by generalizing to tree-structured backbones, enabling even richer representations of latent correlations. Extensive experiments on time-series and graph-structured datasets demonstrate that our proposed method outperforms competitive baselines by effectively capturing higher-order correlations among latent variables, leading to improved performance in downstream tasks.

## 2 Variational Inference with High-order Correlation

### 2.1 Variational Posterior with Instance-Level Correlation Structure

To have the paper focus on its primary objective of capturing instance-level correlations, we assume dimension-level independence in the variational posterior by restricting it to the factorized form $q_\phi(\mathbf{Z}|\mathbf{X}) = \prod_{d=1}^{D} \prod q_\phi([\mathbf{Z}]_{d,1:N}|\mathbf{X})$, where $[\mathbf{Z}]_{d,1:N}$ denotes the $d$-th row of the latent-variable matrix $\mathbf{Z} \in \mathbb{R}^{D \times N}$, although the method can be easily ex-

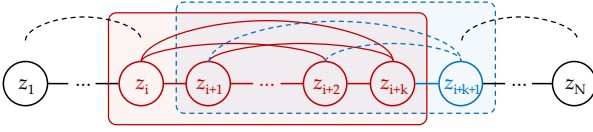

Figure 1: $N$ instances with $k$-order dependency structure.

tended to take the dimension-level correlation into account. To model instance-level correlations, we set the $d$-th variational posterior $q_\phi([\mathbf{Z}]_{d,1:N}|\mathbf{X})$ to be a correlated Gaussian distribution as $q_\phi([\mathbf{Z}]_{d,1:N}|\mathbf{X}) = \mathcal{N}([\mathbf{Z}]_{d,1:N}; \boldsymbol{\mu}_d, \mathbf{P}_d^{-1})$, where $\boldsymbol{\mu}_d \in \mathbb{R}^N$ and $\mathbf{P}_d \in \mathbb{R}^{N \times N}$ denote the mean vector and precision matrix, respectively. By noting that dimensions are handled separately and similarly, in the following, for the conciseness of presentation, we omit the subscript $d$ and observed data $\mathbf{X}$, and simply denote the $d$-th dimensional variational posterior $q_\phi([\mathbf{Z}]_{d,1:N}|\mathbf{X})$ as $q(\mathbf{z}) = \mathcal{N}(\mathbf{z}; \boldsymbol{\mu}, \mathbf{P}^{-1})$, where $\mathbf{z} = [z_1, z_2, \cdots, z_N]^\top$, $\boldsymbol{\mu} = [\mu_1, \mu_2, \cdots, \mu_N]^\top$ and $\mathbf{P} \in \mathbb{R}^{N \times N}$.

Since the instance number $N$ is often very large, which could be as large as tens of thousands or even millions in many scenarios, if we simply set the precision matrix $\mathbf{P}$ as a general matrix, it would be computationally intractable. To balance the computational cost and the capability of modeling high-order correlation, we propose to impose a $k$-order connection structure into the precision matrix $\mathbf{P}$, as shown in Fig. 1, which is equivalent to set the $(i, j)$-th element of $\mathbf{P}$ to zero for any $|i - j| > k$, that is, $p_{ij} = 0$ for any $|i - j| > k$. Note that we here assume the connection structure of latent variables is built upon a chain, which is reasonable for the modeling of sequential data by itself. Later, we will show that the chain backbone is not necessary and can be extended to the more general tree topology to accommodate more diverse data.

Despite the $k$-order connection structure is imposed into $\mathbf{P}$, if we simply substitute the variational posterior $q(\mathbf{z}) = \mathcal{N}(\mathbf{z}; \boldsymbol{\mu}, \mathbf{P}^{-1})$ into the lower bound, we will see that the $N$ latent variables are still coupled together, and we still have to handle them simultaneously. To overcome this issue, we notice that the connection structure is comprised of $N - k$ overlapping local sub-structures, with each involving only $k + 1$ consecutive latent variables $\mathbf{z}_{i:i+k} = [z_i, z_{i+1}, \cdots, z_{i+k}]^\top$ for $i = 1, 2, \cdots, N - k$. The $k + 1$ consecutive variables $\mathbf{z}_{i:i+k}$ follow the marginal distribution of $q(\mathbf{z})$, which can be expressed as

$$q(\mathbf{z}_{i:i+k}) = \mathcal{N}(\mathbf{z}_{i:i+k}; \boldsymbol{\mu}_{i:i+k}, \mathrm{diag}(\boldsymbol{\sigma}_{i:i+k})\mathbf{R}^{(i)}\mathrm{diag}(\boldsymbol{\sigma}_{i:i+k})), \tag{1}$$

where $[\mathbf{R}^{(i)}]_{st} = \gamma_{i+s-1,i+t-1}$ with $[\cdot]_{st}$ denoting the $(s, t)$-th element of a matrix; $\gamma_{i+s-1,i+t-1}$ is the correlation coefficient between latent variables $z_{i+s-1}$ and $z_{i+t-1}$ for any $s, t \in \{1, 2, \cdots, k+1\}$; and $\boldsymbol{\sigma}_{i:i+k} = [\sigma_i, \cdots, \sigma_{i+k}]$, with $\sigma_j$ being the standard deviation of $z_j$. Each of these local marginals $q(\mathbf{z}_{i:i+k})$ represents a localized view of the global variational posterior $q(\mathbf{z})$. Below, we show that the global posterior $q(\mathbf{z})$ can be expressed in terms of these local marginals $q(\mathbf{z}_{i:i+k})$.

**Theorem 2.1.** *For any joint distribution $q(\mathbf{z}) = \mathcal{N}(\mathbf{z}; \boldsymbol{\mu}, \mathbf{P}^{-1})$ with a precision matrix $\mathbf{P}$ that has a $k$-order connection structure, it can be equivalently expressed as*

$$q(\mathbf{z}) = \prod_{i=1}^{N-k+1} q(\mathbf{z}_{i:i+k-1}) \prod_{i=1}^{N-k} \frac{q(\mathbf{z}_{i:i+k})}{q(\mathbf{z}_{i:i+k-1})q(\mathbf{z}_{i+1:i+k})}, \tag{2}$$

*where $q(\mathbf{z}_{i:i+k-1})$ and $q(\mathbf{z}_{i:i+k})$ are the marginals of $q(\mathbf{z})$ over $\mathbf{z}_{i:i+k-1}$ and $\mathbf{z}_{i:i+k}$. Moreover, if the $(k + 1)$-variate marginals $q(\mathbf{z}_{i:i+k})$ are valid distribution for all $i = 1, 2, \cdots, N - k$, then $q(\mathbf{z})$ will also be a valid distribution.*

The theorem reveals that instead of parameterizing the posterior $q(\mathbf{z})$ with $p_{ij}$ in $\mathbf{P}$, we can also use the correlation coefficients $\boldsymbol{\Gamma} \triangleq \{\gamma_{i+s-1,i+t-1} | i = 1, \cdots, N - k, \ |s - t| \leq k, \ s \neq t\}$ to parameterize the local marginals $q(\mathbf{z}_{i:i+k})$ and then use the local marginals to construct the global posterior. From the distribution (2), we can see that the number of parameters in $\boldsymbol{\Gamma}$ is the same as the number of non-zero $p_{ij}$ in the precision matrix $\mathbf{P}$. Thus, without introducing more parameters, the distribution expressed with local marginals enables us to work on $k$-variate local marginals rather than the $N$-variate global posterior, significantly reducing the computational demand.

With the availability of local marginals $q(\mathbf{z}_{i:i+k})$ for $i = 1, 2, \cdots, N - k$, we can use them to draw samples from the high-dimensional global posterior $q(\mathbf{z})$. Specifically, according to the properties of multivariate normal distribution, the conditional distribution of $z_{i+k}$ given $\mathbf{z}_{i:i+k-1} = \tilde{\mathbf{z}}_{i:i+k-1}$ can be expressed as $q(z_{i+k}|\tilde{\mathbf{z}}_{i:i+k-1}) = \mathcal{N}(z_{i+k}; \lambda_{i+k}, \eta_{i+k}^2)$, with the mean and variance equal to

$$\begin{aligned} \lambda_{i+k} &= \mu_{i+k} + [\mathbf{R}^{(i)}]_{k+1,1:k}[\mathbf{R}^{(i)}]_{1:k,1:k}^{-1}(\tilde{\mathbf{z}}_{i:i+k-1} - \boldsymbol{\mu}_{i:i+k-1}) \oslash \boldsymbol{\sigma}_{i:i+k-1}, \\ \eta_{i+k}^2 &= \sigma_{i+k}^2 \left(1 - [\mathbf{R}^{(i)}]_{k+1,1:k}[\mathbf{R}^{(i)}]_{1:k,1:k}^{-1}[\mathbf{R}^{(i)}]_{1:k,k+1}\right), \end{aligned} \tag{3}$$

where $\oslash$ denotes element-wise division; $[\mathbf{A}]_{i:j,s:t}$ means the submatrix of $\mathbf{A}$ with rows from $i$ to $j$ and columns from $s$ to $t$. Thus, given the samples from the $i$-th to the $(i + k - 1)$-th variable $\tilde{\mathbf{z}}_{i:i+k-1} = [\tilde{z}_i, \cdots, \tilde{z}_{i+k-1}]^\top$, the sample drawn from $q(z_{i+k}|\tilde{\mathbf{z}}_{i:i+k-1})$ can be represented as

$$\tilde{z}_{i+k} = \lambda_{i+k}(\boldsymbol{\Gamma}^{(i)}) + \eta_{i+k}(\boldsymbol{\Gamma}^{(i)}) \cdot \epsilon_{i+k}, \quad \epsilon_{i+k} \sim \mathcal{N}(0, 1), \tag{4}$$

where we deliberately write $\lambda_{i+k}$ and $\eta_{i+k}$ as $\lambda_{i+k}(\boldsymbol{\Gamma}^{(i)})$ and $\eta_{i+k}(\boldsymbol{\Gamma}^{(i)})$ to emphasize the sample $\tilde{z}_{i+k}$ is a function of correlation parameters $\boldsymbol{\Gamma}^{(i)} = \{\gamma_{i+s-1,i+t-1} | s, t = 1, \cdots, k + 1, \ s \neq t\}$. With the newly obtained sample $\tilde{z}_{i+k}$ as well as the previous samples $\tilde{\mathbf{z}}_{i+1:i+k-1}$, we can further

draw the next sample $\tilde{z}_{i+k+1}$ from the conditional distribution $q(z_{i+k+1}|\tilde{\mathbf{z}}_{i+1:i+k})$, which can be easily derived from the local marginal $q(\mathbf{z}_{i+1:i+k+1})$. By repeating this process sequentially for $i = 1, 2, \cdots, N - k$, we can obtain the sample $\tilde{\mathbf{z}} = [\tilde{z}_1, \tilde{z}_2, \cdots, \tilde{z}_N]^\top \sim q(\mathbf{z})$.

It should be noted that the sample $\tilde{\mathbf{z}}$ can be explicitly expressed in terms of coefficients $\boldsymbol{\Gamma}$. Thus, we can use the sample $\tilde{\mathbf{z}}$ to estimate the expectation of the evidence lower bound (ELBO) $\mathcal{L}(\boldsymbol{\theta}, \boldsymbol{\phi}, \mathbf{X}) = \mathbb{E}_{\mathbf{z} \sim q_{\boldsymbol{\phi}}(\mathbf{z})}[\log p_{\boldsymbol{\theta}}(\mathbf{X}, \mathbf{z})] + \mathcal{H}[q_{\boldsymbol{\phi}}(\mathbf{z})]$ and give rise to

$$\tilde{\mathcal{L}}(\boldsymbol{\theta}, \boldsymbol{\phi}, \mathbf{X}) = \log p_{\boldsymbol{\theta}}(\mathbf{X}, \tilde{\mathbf{z}}(\boldsymbol{\Gamma})) + \mathcal{H}[q_{\boldsymbol{\phi}}(\mathbf{z})], \tag{5}$$

where the entropy term $\mathcal{H}(\cdot)$ can be expressed in terms of local marginals thanks to the decomposition as depicted in Eq. (2). The exact expression for the ELBO is provided in Appendix C.

To boost inference efficiency, rather than training the coefficients $\boldsymbol{\Gamma}$, it is common to parameterize a neural network $f_{\boldsymbol{\phi}}(\cdot, \cdot)$ to output the coefficient values as $\gamma_{i+s-1,i+t-1} = f_{\boldsymbol{\phi}}(\mathbf{x}_{i+s-1}, \mathbf{x}_{i+t-1})$, where the output value of neural network is confined within the interval $(-1, 1)$ to be consistent with the range of correlation coefficients. By substituting $\gamma_{i+s-1,i+t-1} = f_{\boldsymbol{\phi}}(\mathbf{x}_{i+s-1}, \mathbf{x}_{i+t-1})$ into the lower bound (5), the neural network parameters $\boldsymbol{\phi}$ can be optimized adequately. However, if we directly parameterize $\gamma_{i+s-1,i+t-1}$ as $f_{\boldsymbol{\phi}}(\mathbf{x}_{i+s-1}, \mathbf{x}_{i+t-1})$, the resulting correlation matrix $\mathbf{R}^{(i)}$ could be non-positive definite, which violates the basic requirement of a Gaussian distribution, causing the whole sampling process and ELBO estimation above invalid.

## 2.2 Re-parameterizing the Correlation Matrix $\mathbf{R}^{(i)}$ with Positive Definite Guarantee

To ensure the positive definiteness of $\mathbf{R}^{(i)}$, instead of using neural networks to directly parameterize its elements $\gamma_{i+s-1,i+t-1}$, we propose a new way to parameterize them. Specifically, we notice that for any valid multivariate Gaussian distribution $q(\mathbf{z}_{i:i+k})$, which is equivalent to have $\mathbf{R}^{(i)} \succ 0$, we can always decompose it as

$$q(\mathbf{z}_{i:i+k}) = q(z_i, z_{i+k}|\mathbf{z}_{i+1:i+k-1})q(\mathbf{z}_{i+1:i+k-1}), \tag{6}$$

where the conditional distribution

$$q(z_i, z_{i+k}|\mathbf{z}_{i+1:i+k-1}) = \mathcal{N}\left(\begin{bmatrix} z_i \\ z_{i+k} \end{bmatrix}; \begin{bmatrix} \mu_i^c \\ \mu_{i+k}^c \end{bmatrix}; \begin{bmatrix} \sigma_i^c & 0 \\ 0 & \sigma_{i+k}^c \end{bmatrix} \mathbf{R}_{i,i+k}^c \begin{bmatrix} \sigma_i^c & 0 \\ 0 & \sigma_{i+k}^c \end{bmatrix}\right); \tag{7}$$

$\mu_i^c = \mathbb{E}[z_i|\mathbf{z}_{i+1:i+k-1}]$ and $\mu_{i+k}^c = \mathbb{E}[z_{i+k}|\mathbf{z}_{i+1:i+k-1}]$ are the conditional means; $\sigma_i^c = \mathbb{E}[(z_i - \mu_i^c)^2|\mathbf{z}_{i+1:i+k-1}]^{1/2}$ and $\sigma_{i+k}^c = \mathbb{E}[(z_{i+k} - \mu_{i+k}^c)^2|\mathbf{z}_{i+1:i+k-1}]^{1/2}$ are the conditional standard deviations; and $\mathbf{R}_{i,i+k}^c = \begin{bmatrix} 1 & \gamma_{i,i+k|\mathcal{I}_{i,i+k}}^c \\ \gamma_{i,i+k|\mathcal{I}_{i,i+k}}^c & 1 \end{bmatrix}$ is the conditional correlation matrix. Here, $\gamma_{i,i+k|\mathcal{I}_{i,i+k}}^c$ represents the conditional correlation parameter between $z_i$ and $z_{i+k}$ given $\mathbf{z}_{i+1:i+k-1}$ with index set $\mathcal{I}_{i,i+k} \triangleq \{i+1, \cdots, i+k-1\}$, which can be specifically expressed as

$$\gamma_{i,i+k|\mathcal{I}_{i,i+k}}^c = \frac{\gamma_{i,i+k} - [\mathbf{r}_1^{(i)}]^\top [\mathbf{R}_{k-1}^{(i)}]^{-1} \mathbf{r}_{k+1}^{(i)}}{\sqrt{1 - [\mathbf{r}_1^{(i)}]^\top [\mathbf{R}_{k-1}^{(i)}]^{-1} \mathbf{r}_1^{(i)}} \sqrt{1 - [\mathbf{r}_{k+1}^{(i)}]^\top [\mathbf{R}_{k-1}^{(i)}]^{-1} \mathbf{r}_{k+1}^{(i)}}}, \tag{8}$$

where $\mathbf{r}_1^{(i)} = [\mathbf{R}^{(i)}]_{2:k,1}$, $\mathbf{r}_{k+1}^{(i)} = [\mathbf{R}^{(i)}]_{2:k,k+1}$ and $\mathbf{R}_{k-1}^{(i)} = [\mathbf{R}^{(i)}]_{2:k,2:k}$. For conciseness, we use the notation $\gamma_{i,i+k}^c$ in the following context to represent $\gamma_{i,i+k|\mathcal{I}_{i,i+k}}^c$ without introducing ambiguity. From (8), we can see that there exists a one-to-one mapping $\mathcal{M} : \gamma_{i,i+k}^c \mapsto \gamma_{i,i+k}$ that maps the conditional correlation parameter $\gamma_{i,i+k}^c$ to the correlation parameter $\gamma_{i,i+k}$ in $\mathbf{R}^{(i)}$.

For a valid distribution $q(\mathbf{z}_{i:i+k})$, its conditional distribution $q(z_i, z_{i+k}|\mathbf{z}_{i+1:i+k-1})$ must be valid, too. This suggests that the correlation matrix $\mathbf{R}_{i,i+k}^c$ is positive definite, which is equivalent to the condition $|\gamma_{i,i+k}^c| < 1$. Therefore, for any valid distribution $q(\mathbf{z}_{i:i+k})$, its conditional correlation is ensured to satisfy $|\gamma_{i,i+k}^c| < 1$. Below, we prove that the converse is also true, that is, if we confine $|\gamma_{i,i+k}^c| < 1$ and set $\gamma_{i,i+k} = \mathcal{M}(\gamma_{i,i+k}^c)$, the correlation matrix $\mathbf{R}^{(i)}$ constructed with it is guaranteed to be positive definite under some condition.

**Theorem 2.2.** *By writing the correlation matrix $\mathbf{R}^{(i)}$ as the following partitioned form*

$$\mathbf{R}^{(i)} = \begin{bmatrix} 1 & \gamma_{i,i+1} & \cdots & \gamma_{i,i+k-1} & \gamma_{i,i+k} \\ \gamma_{i,i+1} & 1 & \cdots & \gamma_{i+1,i+k-1} & \gamma_{i+1,i+k} \\ \vdots & \vdots & \ddots & \vdots & \vdots \\ \gamma_{i,i+k-1} & \gamma_{i+1,i+k-1} & \cdots & 1 & \gamma_{i+k-1,i+k} \\ \gamma_{i,i+k} & \gamma_{i+1,i+k} & \cdots & \gamma_{i+k-1,i+k} & 1 \end{bmatrix}, \tag{9}$$

*if the upper-left and lower-right sub-matrices $[\mathbf{R}^{(i)}]_{1:k,1:k}$ and $[\mathbf{R}^{(i)}]_{2:k+1,2:k+1}$ in the dotted frames are both positive definite, $|\gamma_{i,i+k}^c| < 1$ and we set $\gamma_{i,i+k} = \mathcal{M}(\gamma_{i,i+k}^c)$, then $\mathbf{R}^{(i)}$ is positive definite.*

According to Theorem 2.2, if $k \times k$ sub-matrices $[\mathbf{R}^{(i)}]_{1:k,1:k}$ and $[\mathbf{R}^{(i)}]_{2:k+1,2:k+1}$ of the $(k+1) \times (k+1)$ correlation matrix $\mathbf{R}^{(i)}$ are both positive definite, and we let $\gamma_{i,i+k} = \mathcal{M}(\gamma_{i,i+k}^c)$ with $|\gamma_{i,i+k}^c| < 1$, then the correlation matrix $\mathbf{R}^{(i)}$ constructed in the form of (9) is guaranteed to be positive definite. This gives rise to an iterative construction approach, starting from small sub-matrices and expanding step by step. To illustrate this process, let us take the construction of $4 \times 4$ correlation matrix as an example, whose eventual form is

$$\mathbf{R}^{(1)} = \begin{bmatrix} 1 & \gamma_{12} & \mathcal{M}(\gamma_{13|2}^c) & \mathcal{M}(\gamma_{14|23}^c) \\ \gamma_{12} & 1 & \gamma_{23} & \mathcal{M}(\gamma_{24|3}^c) \\ \mathcal{M}(\gamma_{13|2}^c) & \gamma_{23} & 1 & \gamma_{34} \\ \mathcal{M}(\gamma_{14|23}^c) & \mathcal{M}(\gamma_{24|3}^c) & \gamma_{34} & 1 \end{bmatrix}. \tag{10}$$

If $\gamma_{12} < 1$ and $\gamma_{23} < 1$, sub-matrices $[\mathbf{R}^{(1)}]_{1:2,1:2}$ and $[\mathbf{R}^{(1)}]_{2:3,2:3}$ are known to be positive definite. Then, if we confine $\gamma_{13|2}^c < 1$, according to Theorem 2.2, the sub-matrix $[\mathbf{R}^{(1)}]_{1:3,1:3}$ is ensured to be positive definite. Similarly, if $\gamma_{23}, \gamma_{34}, \gamma_{24|3}^c$ lie within $(-1, 1)$, we can also ensure $[\mathbf{R}^{(1)}]_{2:4,2:4} \succ 0$. Then, combining with the condition $\gamma_{24|23}^c < 1$, we can see from Theorem 2.2 that the correlation matrix $\mathbf{R}^{(1)}$ is guaranteed to be positive definite. Continued in this way recursively, positive definite correlation matrices of arbitrary size can be constructed, as depicted in the following corollary.

**Corollary 2.3.** *If all correlation parameters in $\mathbf{\Gamma}_1 = \{\gamma_{i,i+1}\}_{i=1}^{N-1}$ and $\mathbf{\Gamma}_t = \{\gamma_{i,i+t}^c\}_{i=1}^{N-t}$ for $t = 2, 3, \cdots, k$ lie in the interval $(-1, 1)$, then the $(k+1) \times (k+1)$ correlation matrix $\mathbf{R}^{(i)}$ constructed as above is guaranteed to be positive definite.*

Therefore, to construct a correlation matrix $\mathbf{R}^{(i)}$ with positive definite guarantee, we only need to parameterize first-order correlations $\mathbf{\Gamma}_1$ and higher-order conditional correlations $\mathbf{\Gamma}_t$ for $t = 2, 3, \cdots, k$, and ensure them to lie in the interval $(-1, 1)$. For different orders of correlation coefficients, we can use a specific neural network $f_{\phi_t}(\cdot, \cdot)$ to parameterize them as

$$\begin{aligned} \gamma_{i,i+1} &= f_{\phi_1}(\mathbf{x}_i, \mathbf{x}_{i+1}), \quad i = 1, 2, \cdots, N-1, \\ \gamma_{i,i+t}^c &= f_{\phi_t}(\mathbf{x}_i, \mathbf{x}_{i+t}), \quad i = 1, 2, \cdots, N-t, \end{aligned} \tag{11}$$

which represent the first-order correlations and $t$-order conditional correlations, respectively. Once the positive definite correlation matrix $\mathbf{R}^{(i)}$ has been constructed, we can then use the method described in Section 2.1 to optimize the ELBO in (5) safely.

The exact cost of inference with our proposed k-order precision matrix $\mathbf{P}$ involves three parts: (i) the cost of neural network evaluations for re-parameterizing correlation coefficients, (ii) the cost of sampling from the variational posterior, and (iii) entropy calculation. In our method, to define a posterior with k-order correlation over $N$ latent variables, we need to specify exactly $(N-1)$ first-order, $(N-2)$ second-order, and so on, up to $(N-k)$ k-order correlations, yielding a total of $(N-1) + (N-2) + \cdots + (N-k) = k(2N-k-1)/2$ correlation coefficients. In our method, each coefficient is parameterized by the output of a re-parameterization network $f_\phi(\cdot, \cdot)$. Therefore, to re-parameterize these coefficients, we need to run the network $f_\phi(\cdot, \cdot)$ for $\mathcal{O}(kN)$ times. The sampling and entropy calculations involve operations like inversion in (3) and determinant computation on $k \times k$ sub-matrices, incurring a cost of $\mathcal{O}(k^3)$ FLOPs. Considering that $k$ is typically much smaller than $N$ and the complexity of evaluating neural networks, the cost of these operations is negligible compared to the cost of neural network evaluations. Therefore, the total cost approximately amounts to the cost of evaluating $\mathcal{O}(kN)$ times of the neural network $f_\phi$ per epoch, which is approximately $k$ times of the cost of mean-field amortized VI methods, with the order $k$ controlling the trade-off between expressiveness and computational cost.

## 2.3 Extensions to Tree-structured Backbones

Although Theorem 2.1 is built on a chain-structured backbone, our proposed high-order correlations could be extended to more general tree-structured backbones. If the first-order dependency structure among latent variables is characterized by a tree-structured backbone as shown by the solid lines in Figure 2, then we can also impose $k$-order dependencies over the tree-structured backbone. In this case, every $k+1$ consecutive variables on the tree forms a

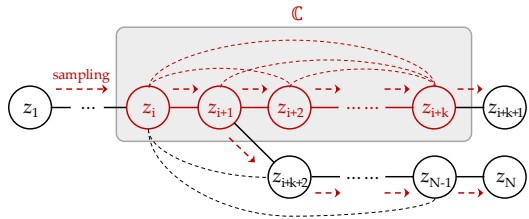

Figure 2: $N$ instances with $k$-order dependency structure based on a tree-structured backbone.

$(k+1)$-vertex clique $\mathbb{C} \in \mathcal{C}_{k+1}$ with indices $\mathbb{C} = \{i_0, i_1, \cdots, i_k\} \subseteq \{1, 2, \cdots, N\}$. Based on the tree-structured backbone, we can extend Theorem 2.1 to equivalently express the joint distribution of latent variables using its local marginals over $k$-vertex cliques $\mathcal{C}_k$ and $(k+1)$-vertex cliques $\mathcal{C}_{k+1}$ as follows

$$q(\mathbf{z}) = \prod_{\mathbb{C}=\{i_1,\cdots,i_k\}\in\mathcal{C}_k} q(z_{i_1}, \cdots, z_{i_k}) \prod_{\substack{\mathbb{C}=\{i_0,i_1,\cdots,i_k\}\in\mathcal{C}_{k+1} \\ i_0 < i_1 < \cdots < i_k}} \frac{q(z_{i_0}, z_{i_1}, \cdots, z_{i_{k-1}}, z_{i_k})}{q(z_{i_0}, \cdots, z_{i_{k-1}})q(z_{i_1}, \cdots, z_{i_k})}, \quad (12)$$

which is fully determined by local marginals $q(z_{i_0}, \cdots, z_{i_k})$ over $(k+1)$-vertex cliques $\mathbb{C} = \{i_0, i_1, \cdots, i_k\} \in \mathcal{C}_{k+1}$. The validity of the local marginals can be similarly guaranteed by parameterizing the correlation matrices with first-order correlations and higher-order conditional correlations, and further confining them within $(-1, 1)$, as the following corollary shows.

**Corollary 2.4.** *If the first-order correlations $\mathbf{\Gamma}_1$ and higher-order conditional correlations $\mathbf{\Gamma}_t$ for $t = 2, 3, \cdots, k$ are built upon a tree-structured backbone, and all correlation parameters lie in the interval $(-1, 1)$, then we can use them to construct a $(k+1) \times (k+1)$ correlation matrix $\mathbf{R}^{(i)}$ with $k$-order dependency structure.*

Given the marginal distribution and samples $z_{i_0}, \cdots, z_{i_{k-1}}$, we can draw sample $z_{i_k}$ from the conditional distribution $q(z_{i_k}|z_{i_0}, \cdots, z_{i_{k-1}})$. By recursively sampling from the conditional distribution starting from the root node, samples from the joint distribution can be obtained.

## 3 Related Work

Bayesian inference provides a principled framework for uncertainty estimation, but exact inference is often intractable. Variational inference addresses this by approximating the true posterior with a more tractable distribution. This requires a trade-off between expressiveness and computational efficiency [8]. A widely-used approach is mean-field variational inference (MFVI) [11], which assumes a fully factorized posterior, treating all latent variables as independent. Despite its broad applicability across domains such as image analysis [57] and biology [2], MFVI struggles to capture posterior correlations, particularly in settings where latent variables are strongly dependent. To address these limitations, structured variational inference (SVI) enriches variational distributions to capture dependencies among latent variables while retaining tractability. Common SVI approaches achieve this through deterministic or stochastic transformations, such as normalizing flows [10, 61] and implicit models [54, 41]. Other techniques include modeling local-global dependencies [22, 60], using mixture distributions [42, 34], copula-based augmentations [26, 53], non-conjugacy approximations [28, 49], and hierarchical extensions [1, 39]. While these methods enhance expressiveness, they primarily focus on intra-instance correlations, limiting their scalability to capturing correlations across instances. Another related thread of work is neural relational inference [30, 14], which models the latent interactions among entities or objects across data points using graph-based representations. While effective in discovering relational structures, these methods focus on structure learning and do not explicitly leverage inter-instance dependencies to enhance the variational approximation itself.

Higher-order dependencies have emerged as a crucial modeling component in complex systems where first-order representations fall short [46]. These dependencies, which account for interactions involving three or more entities, are prevalent in real-world sequential data such as multivariate time series, clickstreams [65], citation flows [23], and transportation systems [65]. To capture these higher-order interactions, several modeling paradigms have been developed, including hypergraphs

[32, 9], simplicial complexes [13, 5], motif-based networks [7, 6], and higher-order Markov models [52, 18]. However, these methods often suffer from scalability issues due to the exponential growth of dependencies. Recent efforts have aimed to incorporate instance-level dependencies into variational inference [43, 38, 56, 63]. For instance, DC-GMM [38] introduces a prior information matrix to promote similar posteriors across instances for weakly supervised clustering. However, it still relies on a mean-field approximation, which limits its ability to fully capture correlated posteriors. Other methods, such as CVAE [56], attempt to address this limitation by constructing tree-structured variational posteriors, effectively modeling pairwise dependencies among instances. But they remain limited to first-order correlations and struggle to represent higher-order dependencies. The work of TreeVI [63] is most similar to ours, but is inherently limited to modeling only first-order correlations. This limitation of TreeVI arises from its reliance on an acyclic tree structure to construct its correlation matrix. While this is sufficient for simple pairwise relationships, attempting to model higher-order correlations inherently introduces loops into the underlying correlation structure. The construction of TreeVI depends on the acyclic property of its backbone and is no longer valid when these loops exist. Moreover, simply modeling higher-order correlation coefficients within the framework of TreeVI does not guarantee the correlation matrix to be positive definite. So even though TreeVI can capture instance-level correlations, it cannot be easily extended to model higher-order correlations. In addition, SIDEC [58] takes a different approach by leveraging variational inference to learn latent dynamics and employing high-order correlations for structural reconstruction. However, its focus is on recovering interaction graphs rather than explicitly modeling high-order dependencies in the latent posteriors themselves.

## 4   Experiments

**Tasks & Datasets.**   We evaluate our method on three tasks: time series anomaly detection, time series forecasting and constrained clustering, using a diverse set of benchmark datasets. For time series anomaly detection, we experiment on three datasets: SMD, SMAP, and MSL. For time series forecasting, we use five widely-used datasets: ETTh1, ETTm1, Electricity, Exchange, and Weather. For constrained clustering, we conduct experiments on four standard datasets: MNIST, Fashion MNIST, Reuters, and STL-10. Detailed descriptions for each dataset are provided in Appendix D.1.

**Baselines & Implementation Details.**   For time series anomaly detection, we compare our method with four state-of-the-art unsupervised approaches for time series anomaly detection based on VAE: DAGMM [70], LSTM-VAE [45], OmniAnomaly [55], and SISVAE [37]. For time series forecasting, we compare our method with the state-of-the-art end-to-end methods on time series modeling and forecasting tasks, including VRAE [20], Informer [69], GRU-NVP [47], and DeepAR [50]. For constrained clustering, we compare our approach against traditional algorithms such as PCKMeans [4], SDEC [48], C-IDEC [67], and the state-of-the-art DC-GMM [38]; and also benchmark against generative models such as VaDE [27], DGG [66], and TreeVI [63]. To evaluate the effectiveness of our method, we conduct experiments with varying levels of $k$-order dependency structures, specifically using $k \in \{1, 3, 5, 10\}$. Further implementation details are provided in Appendix D.3.

### 4.1   Time Series Modeling

Generative Time Series Modeling aims to learn the underlying probability distribution of time series data and generate new, synthetic time series samples that exhibit similar characteristics to the observed data. However, the majority of existing approaches often ignore instance-level correlations during posterior inference, thus failing to comprehensively cap-

Table 2: F1-Score and Evidence Lower Bound comparisons.

| Dataset | | SMAP | | MSL | | SMD | |
|---------|---------|--------|----------|--------|----------|--------|----------|
| Metric | | F1 | ELBO | F1 | ELBO | F1 | ELBO |
| DAGMM | | 0.7105 | -115.2820 | 0.7007 | -277.7380 | 0.7094 | -155.9460 |
| LSTM-VAE | | 0.7298 | -116.9500 | 0.6780 | -281.3220 | 0.7842 | -146.0540 |
| OmniAnomaly | | 0.8434 | -98.9217 | 0.8849 | -161.0002 | 0.8857 | -72.0419 |
| SISVAE | | 0.8299 | -101.1878 | 0.8766 | -182.6060 | 0.8775 | -72.5832 |
| HoT-VI | 1-order | 0.8411 | -97.6057 | 0.8883 | -165.5004 | 0.8901 | -69.5278 |
| | 3-order | 0.8552 | -95.2314 | 0.8940 | -157.2134 | 0.9153 | -67.4001 |
| | 10-order | **0.8636** | **-92.2948** | **0.9145** | **-134.0815** | **0.9284** | **-65.0345** |

ture the temporal dynamics of time series. To address this limitation, our method incorporates two key adaptations compared to the vanilla VAE. First, temporal dependencies are introduced by integrating Gated Recurrent Units (GRUs) [15] in both the VAE encoder and decoder. Second, the

Table 1: Multivariate time series forecasting results with horizon $H \in \{24, 48, 168, 336, 720\}$. Best performance is highlighted in bold font and the second best results are underlined.

| Method | | Informer | | GRU-NVP | | DeepAR | | VRAE | | HoT-VI (Ours) | | | | | |
| | | | | | | | | | | 1-order | | 3-order | | 10-order | |
| | | MSE | MAE | MSE | MAE | MSE | MAE | MSE | MAE | MSE | MAE | MSE | MAE | MSE | MAE |
|---|---|---|---|---|---|---|---|---|---|---|---|---|---|---|---|
| ETTh1 | 24 | 0.577 | 0.549 | 3.540 | 0.733 | 1.166 | 0.836 | 0.743 | 0.762 | 0.664 | 0.570 | 0.543 | 0.505 | **0.363** | **0.376** |
| | 48 | 0.685 | 0.625 | 2.549 | 0.622 | 1.154 | 0.827 | 0.826 | 0.801 | 0.705 | 0.597 | 0.578 | 0.528 | **0.392** | **0.392** |
| | 168 | 0.931 | 0.752 | 3.831 | 0.774 | 1.083 | 0.778 | 1.070 | 0.938 | 0.848 | 0.681 | 0.721 | 0.615 | **0.510** | **0.464** |
| | 336 | 1.128 | 0.873 | 6.877 | 1.008 | 1.043 | 0.766 | 1.199 | 1.016 | 0.990 | 0.755 | 0.883 | 0.702 | **0.616** | **0.525** |
| | 720 | 1.215 | 0.896 | 5.377 | 1.060 | 1.075 | 0.795 | 1.426 | 1.164 | 1.129 | 0.821 | 1.021 | 0.781 | **0.763** | **0.630** |
| ETTm1 | 24 | 0.453 | 0.444 | 0.605 | 0.437 | 1.360 | 0.871 | 0.687 | 0.646 | 0.488 | 0.455 | 0.409 | 0.417 | **0.253** | **0.298** |
| | 48 | 0.494 | 0.503 | 2.787 | 0.701 | 1.334 | 0.866 | 0.817 | 0.724 | 0.648 | 0.544 | 0.535 | 0.488 | **0.330** | **0.345** |
| | 168 | 0.678 | 0.614 | 4.212 | 0.824 | 1.170 | 0.838 | 0.853 | 0.794 | 0.686 | 0.573 | 0.578 | 0.521 | **0.368** | **0.373** |
| | 336 | 1.056 | 0.786 | 5.062 | 1.019 | 1.249 | 0.846 | 1.091 | 0.975 | 0.771 | 0.628 | 0.641 | 0.567 | **0.434** | **0.415** |
| | 720 | 1.192 | 0.926 | 5.799 | 1.075 | 1.075 | 0.770 | 1.165 | 0.996 | 0.886 | 0.692 | 0.737 | 0.626 | **0.528** | **0.474** |
| Electricity | 24 | 0.312 | 0.387 | 3.514 | 1.844 | 0.211 | 0.330 | 0.279 | 0.396 | 0.326 | 0.400 | 0.256 | 0.346 | **0.134** | **0.238** |
| | 48 | 0.392 | 0.431 | 3.318 | 1.786 | 0.332 | 0.398 | 0.317 | 0.410 | 0.347 | 0.415 | 0.277 | 0.363 | **0.152** | **0.255** |
| | 168 | 0.515 | 0.509 | 3.482 | 1.833 | 1.065 | 0.811 | 0.366 | 0.475 | 0.373 | 0.433 | 0.303 | 0.382 | **0.174** | **0.273** |
| | 336 | 0.759 | 0.625 | 3.921 | 1.941 | 1.040 | 0.795 | 0.402 | 0.515 | 0.388 | 0.445 | 0.319 | 0.395 | **0.194** | **0.293** |
| | 720 | 0.969 | 0.788 | 4.232 | 2.020 | 1.048 | 0.804 | 0.450 | 0.556 | 0.415 | 0.463 | 0.348 | 0.416 | **0.230** | **0.323** |
| Exchange | 24 | 0.611 | 0.626 | 1.557 | 0.877 | 1.328 | 0.692 | 0.140 | 0.310 | 0.098 | 0.227 | 0.093 | 0.227 | **0.033** | **0.126** |
| | 48 | 0.680 | 0.644 | 1.589 | 0.883 | 1.345 | 0.701 | 0.238 | 0.435 | 0.155 | 0.267 | 0.171 | 0.306 | **0.058** | **0.164** |
| | 168 | 1.097 | 0.825 | 1.663 | 0.903 | 1.434 | 0.745 | 0.642 | 0.703 | 0.379 | 0.466 | 0.368 | 0.458 | **0.196** | **0.326** |
| | 336 | 1.672 | 1.036 | 1.682 | 0.905 | 1.489 | 0.778 | 1.050 | 0.953 | 0.992 | 0.835 | 1.165 | 0.821 | **0.496** | **0.515** |
| | 720 | 2.478 | 1.310 | 1.748 | 0.928 | 1.526 | **0.793** | 3.003 | 1.593 | 1.988 | 1.063 | 2.029 | 1.090 | **1.508** | 0.857 |
| Weather | 24 | 0.162 | 0.235 | 1.222 | 0.909 | 0.205 | 0.250 | 0.227 | 0.315 | 0.206 | 0.294 | 0.186 | 0.281 | **0.129** | **0.179** |
| | 48 | 0.348 | 0.400 | 2.319 | 1.287 | 0.229 | 0.267 | 0.449 | 0.495 | 0.325 | 0.385 | 0.291 | 0.361 | **0.186** | **0.230** |
| | 168 | 0.444 | 0.463 | 2.174 | 1.165 | 0.344 | 0.343 | 0.563 | 0.648 | 0.466 | 0.506 | 0.429 | 0.486 | **0.294** | **0.313** |
| | 336 | 0.578 | 0.523 | 2.119 | 1.221 | 0.568 | 0.527 | 0.781 | 0.841 | 0.767 | 0.645 | 0.625 | 0.575 | **0.550** | **0.430** |
| | 720 | 1.059 | 0.741 | 2.621 | 1.303 | **0.571** | 0.533 | 1.125 | 1.058 | 0.998 | 0.727 | 0.808 | 0.653 | 0.772 | **0.510** |
| Average | | 0.819 | 0.660 | 3.112 | 1.122 | 0.978 | 0.678 | 0.796 | 0.741 | 0.641 | 0.555 | 0.573 | 0.516 | **0.386** | **0.373** |

posterior is approximated using $k$-order dependency. The formal representations are as follows. Let $\mathbf{X} = \{\mathbf{x}_t\}_{t=1}^N$ denote a time series comprising $N$ observations, where each $\mathbf{x}_t \in \mathbb{R}^C$ represents observations across $C$ channels at time step $t$. Given a window of $T$ observations $\mathbf{X}_{t-T+1:t}$, the encoder is represented as: $\boldsymbol{\mu}, \boldsymbol{\sigma} = f_\phi([\mathbf{e}_{t-T}, ..., \mathbf{e}_t])$, where the hidden state of GRU encoder $\mathbf{e}_t$ is updated by $\mathbf{x}_t$ and $\mathbf{e}_{t-1}$. Similarly, the decoder is represented as: $\mathbf{X}_{t-T:t} = f_\theta([\mathbf{d}_{t-T+1}, ..., \mathbf{d}_t])$, where $\mathbf{d}_t$ is the hidden state of the GRU decoder and updated by $\mathbf{z}_t$ and $\mathbf{d}_{t-1}$. We evaluate the modeling capacity of our method through its performance on two downstream tasks.

**Time Series Anomaly Detection** The objective of this task is to determine whether an observation $\mathbf{x}_t$ is anomalous based on the preceding $T$ observations. Our model can be directly applied to the anomaly detection task by reconstructing data. Trained solely on normal data, the model is expected to exhibit low reconstruction loss for normal data while high for anomalies. Consequently, anomalies are identified by comparing the reconstruction loss against a threshold. The ELBO serves as a metric to evaluate the modeling capacity for normal data, while the F1 score assesses anomaly detection performance. As shown in Table 2, our method demonstrates superior ELBO and F1 scores compared to other generative approaches that neglect instance-level correlations during posterior inference. Notably, even with only first-order dependencies, our method achieves comparable performance to OmniAnomaly, a complex model integrating VAE, flow, and State Space Models (SSM). Furthermore, increasing the order of dependencies in our model leads to consistently higher ELBO and F1 scores than all baselines. This indicates that modeling higher-order temporal relationships in time series improves data modeling and anomaly detection performance. By modeling $k$-order dependency, our model captures fine-grained local dynamics and coarser-grained long-term dependency, leading to a more robust and comprehensive understanding of the complex temporal structure.

**Time Series Forecasting** This task aims to predict the subsequent $H$ observations given $L$ past observations. Formally, this is a mapping $f : \mathbf{X}_{t-L+1:t} \in \mathbb{R}^{L \times C} \mapsto \bar{\mathbf{X}}_{t+1:t+H} = \mathbf{Y} \in \mathbb{R}^{H \times C}$, and we omit the subscripts hereafter. Our approach is decomposed into two steps: first, learn an expressive and predictable representation of the historical observations via generative modeling; second, perform forecasting based on the representation. For generative modeling, we capture instance-level correlations using $k$-order dependency that existing approaches often overlook. For forecasting, we integrate a feed forward network $f_\psi : \mathbf{Z} \mapsto \mathbf{Y}$ into the original model. Formally, we aim to optimize the

Table 3: Clustering performances (%) of our proposed method compared with baselines. Means and standard deviations are computed across 10 runs with different random initializations.

| Dataset | Metric | VaDE | SDEC | C-IDEC | DGG | DC-GMM | TreeVI | HoT-VI (*Ours*) | | |
| --- | --- | --- | --- | --- | --- | --- | --- | --- | --- | --- |
| | | | | | | | | 3-order | 5-order | 10-order |
| MNIST | ACC | 89.0±5.0 | 86.2±0.1 | 96.3±0.2 | 95.8±0.1 | 96.5±0.2 | 97.4±0.3 | **98.1±0.4** | **98.3±0.4** | **98.5±0.3** |
| | NMI | 82.8±3.0 | 84.2±0.1 | 91.8±1.0 | 91.2±0.2 | 91.4±0.3 | 93.1±0.6 | **93.8±0.4** | **94.2±0.3** | **94.6±0.3** |
| | ARI | 80.9±5.0 | 80.1±0.1 | 92.1±0.4 | 91.4±0.3 | 92.5±0.5 | 93.7±0.7 | **94.9±0.6** | **95.2±0.5** | **95.6±0.5** |
| fMNIST | ACC | 55.1±2.2 | 54.0±0.2 | 68.1±3.0 | 79.9±0.4 | 80.5±0.8 | 81.4±0.6 | **82.9±0.5** | **83.2±0.5** | **83.4±0.4** |
| | NMI | 57.9±2.7 | 57.3±0.1 | 66.7±2.0 | 70.1±0.3 | 72.0±0.4 | 73.9±0.6 | **74.7±0.6** | **74.8±0.5** | **75.1±0.4** |
| | ARI | 41.6±3.1 | 40.2±0.1 | 52.3±3.0 | 64.9±0.3 | 66.4±0.5 | 67.9±0.9 | **68.9±0.5** | **69.1±0.5** | **69.2±0.4** |
| Reuters | ACC | 76.0±0.7 | 82.1±0.1 | 94.7±0.6 | 93.5±0.6 | 95.4±0.2 | 95.9±0.6 | **96.8±0.6** | **97.2±0.6** | **97.6±0.5** |
| | NMI | 50.1±1.3 | 62.3±0.1 | 81.4±0.7 | 81.2±0.8 | 82.7±0.7 | 83.4±0.5 | **84.8±0.6** | **85.1±0.6** | **85.4±0.5** |
| | ARI | 58.0±1.4 | 66.7±0.1 | 87.7±0.9 | 87.8±0.5 | 89.0±0.6 | 90.2±0.4 | **91.3±0.5** | **91.6±0.5** | **92.0±0.4** |
| STL-10 | ACC | 77.3±0.5 | 79.2±0.1 | 81.6±3.8 | 89.9±0.3 | 89.5±0.5 | 90.4±0.9 | **91.8±0.7** | **92.2±0.6** | **92.4±0.4** |
| | NMI | 70.6±0.4 | 78.6±0.1 | 77.3±1.7 | 80.9±0.5 | 80.2±0.7 | 81.3±0.8 | **82.4±0.7** | **82.8±0.6** | **83.1±0.5** |
| | ARI | 62.7±0.4 | 71.0±0.1 | 71.8±3.4 | 79.0±0.4 | 78.4±0.9 | 79.5±0.7 | **80.9±0.7** | **81.3±0.5** | **81.5±0.5** |

joint model $p(\mathbf{X}, \mathbf{Y}) = p(\mathbf{Y}|\mathbf{X})p(\mathbf{X}) = \int_{\mathbf{Z}} p(\mathbf{Y}|\mathbf{Z})p(\mathbf{Z}|\mathbf{X})d\mathbf{Z} \int_{\mathbf{Z}} p(\mathbf{X}|\mathbf{Z})p(\mathbf{Z})d\mathbf{Z}$ by maximizing the evidence lower bound: $\log p(\mathbf{X}, \mathbf{Y}) \geq \log \int_{\mathbf{Z}} p_\psi(\mathbf{Y}|\mathbf{Z})q_\phi(\mathbf{Z}|\mathbf{X})d\mathbf{Z} + \mathbb{E}_{q_\phi(\mathbf{Z}|\mathbf{X})}\left[\log p_\theta(\mathbf{X}|\mathbf{Z})\right] - \mathbb{KL}\left(q_\phi(\mathbf{Z}|\mathbf{X})\|p(\mathbf{Z})\right)$, where the first term measures prediction accuracy, typically estimated using $L_1$ or $L_2$ loss, while the subsequent terms serve as regularization for representation learning. The results presented in Table 1 show that our approach outperforms all baselines across all datasets and metrics with only two second-best exceptions. Furthermore, the consistent prediction improvement in our method with increasing dependency order underscores that time series forecasting can benefit from better time series modeling.

## 4.2 Constrained Clustering

Constrained clustering is a task that incorporates instance-level constraints into the clustering process, allowing users to enforce specific relationships between data points based on prior knowledge. These constraints are expressed by a correlation graph $\mathbb{G} = (\mathcal{V}, \mathcal{E}, \mathbf{A})$, where $\mathcal{V}$ denotes the set of instances, and the edge set $\mathcal{E} = \mathcal{E}_M \cup \mathcal{E}_C$ consists of *must-link* constraints $\mathcal{E}_M$, requiring two instances to be in the same cluster, and *cannot-link* constraints $\mathcal{E}_C$, which require them to be in different clusters. The adjacency matrix $\mathbf{A} \in \mathbb{R}^{N \times N}$ encodes both the type and strength of each constraint: $[\mathbf{A}]_{ij} > 0$ if $(i, j) \in \mathcal{E}_M$, $[\mathbf{A}]_{ij} < 0$ if $(i, j) \in \mathcal{E}_C$, and $[\mathbf{A}]_{ij} = 0$ if no constraint exists. The magnitude $|[\mathbf{A}]|_{ij} \in [0, \infty)$ reflects the confidence in the constraint. Following the generative modeling framework of previous work [38], constrained clustering can be formulated as a probabilistic clustering problem with joint probability $p_\theta(\mathbf{X}, \mathbf{Z}, \mathbf{c}|\mathbf{A}) = p_\theta(\mathbf{X}|\mathbf{Z})p(\mathbf{Z}|\mathbf{c})p(\mathbf{c}|\mathbf{A})$, where the data $\mathbf{x}_i$ is generated from a normal distribution conditioned on $\mathbf{z}_i$; the latent embedding $\mathbf{z}_i$ is drawn from a cluster-dependent normal distribution $p(\mathbf{z}_i|c_i) = \mathcal{N}(\mathbf{z}_i; \boldsymbol{\mu}_{c_i}, \text{diag}(\boldsymbol{\sigma}_{c_i}^2))$; and the cluster assignments $\mathbf{c} = \{c_i\}_{i=1}^N$ follow a distribution conditioned on $\mathbf{A}$, defined as $p(\mathbf{c}|\mathbf{A}) = \frac{1}{\Omega(\boldsymbol{\pi})} \prod_i \pi_{c_i} h_i(\mathbf{c}, \mathbf{A})$, where $h_i(\mathbf{c}, \mathbf{A}) = \prod_{j \neq i} \exp([\mathbf{A}]_{ij}\delta_{c_i c_j})$ is a weighting function with $\delta$ representing the indicator function, $\boldsymbol{\pi}$ are the cluster weights, and $\Omega(\boldsymbol{\pi}) = \sum_{\mathbf{c}} \prod_i \pi_{c_i} h_i(\mathbf{c}, \mathbf{A})$ is a normalization constant.

To perform inference, we use a variational posterior of the form $q_\phi(\mathbf{Z}, \mathbf{c}|\mathbf{X}) = q_\phi(\mathbf{Z}|\mathbf{X})q(\mathbf{c}|\mathbf{Z})$, where $q(\mathbf{c}|\mathbf{Z}) = \prod_i q(c_i|\mathbf{z}_i)$ is computed using Bayes' rule. In standard approaches like DC-GMM [38], the posterior $q_\phi(\mathbf{Z}|\mathbf{X})$ is modeled as fully factorized, which ignores dependencies between instances. We address this limitation by introducing a higher-order dependency structure over the latent space. Specifically, we approximate $q_\phi(\mathbf{Z}|\mathbf{X})$ using $k$-order correlations, where first-order dependencies are guided by a tree structure learned from the correlation graph $\mathbb{G}$. We follow the work of [63] to learn the tree structure from data by optimizing a symmetric adjacency matrix. In our experiments, we set $k \in \{3, 5, 10\}$ and compare our model with baselines over 10 independent runs, reporting average Accuracy (ACC), Normalized Mutual Information (NMI), and Adjusted Rand Index (ARI) in Table 3. The results show that our approach outperforms existing methods across all datasets and metrics, demonstrating the effectiveness of incorporating higher-order correlations in constrained clustering. The averaged improvements of our method incorporating third-order dependency structure are 1.93, 2.35 and 2.43 in ACC, NMI and ARI against DC-GMM and are 1.13, 1.00 and 1.18 against TreeVI, underscoring the significance of considering dependencies among latent

Table 4: Additional experiments of HoT-VI with orders $k$ exceeding 10, including time series anomaly detection on SMAP dataset, time series forecasting on ETTh1 dataset with horizon 24, and constrained clustering on MNIST dataset.

| Methods | Mean-field | $k=1$ | $k=3$ | $k=10$ | $k=50$ | $k=100$ |
|---|---|---|---|---|---|---|
| *Time Series Anomaly Detection* | | | | | | |
| Runtime (s) | 1.00 | 2.44 | 8.51 | 27.60 | 142.58 | 294.53 |
| F1 | 0.7774 | 0.8411 | 0.8552 | 0.8636 | 0.8711 | 0.8755 |
| ELBO | -109.2182 | -97.6057 | -95.2314 | -92.2948 | -90.4291 | -89.9577 |
| *Time Series Forecasting* | | | | | | |
| Runtime (s) | 4.80 | 11.45 | 36.94 | 126.44 | 675.75 | 1340.22 |
| MSE | 0.739 | 0.664 | 0.543 | 0.363 | 0.348 | 0.333 |
| MAE | 0.716 | 0.570 | 0.505 | 0.376 | 0.362 | 0.352 |
| *Constrained Clustering* | | | | | | |
| Runtime (s) | 0.25 | 0.59 | 1.84 | 6.22 | 29.53 | 60.59 |
| ACC (%) | 96.50 | 97.55 | 98.12 | 98.52 | 98.62 | 98.69 |
| NMI (%) | 91.37 | 93.44 | 93.80 | 94.55 | 94.63 | 94.85 |
| ARI (%) | 92.54 | 93.89 | 94.89 | 95.65 | 95.85 | 96.09 |

posteriors, particularly higher-order dependencies. Furthermore, the performance of our method consistently improves with increasing dependency order, benefiting from the ability of higher-order correlations to jointly link a larger set of data instances. This facilitates more effective propagation of cluster assignment constraints compared to methods limited to pairwise dependencies, further underscoring the importance of capturing high-order interactions in constrained clustering.

**Performances At Higher Orders** The choice of the order $k$ is a trade-off between model expressiveness and computational cost. Generally, as $k$ increases, the model's performance consistently improves, as demonstrated in our experimental results. However, as seen from Table 4, the performance gains diminish as the order $k$ (e.g. 50, 100) goes higher. However, the computational cost always scales linearly with $k$. To balance the gains and cost, we set $k$ to moderate values (up to 10) in our main experiments. By setting $k$ to a moderate value (e.g. 10), we can only model correlation up to 10-th order, losing the ability to model higher-order correlations. But as observed from Table 4, the gains become increasingly weak as the order goes higher.

## 5 Conclusion

In this work, we introduced a novel variational inference framework for modeling higher-order correlations among latent variables, going beyond the limitations of mean-field and first-order methods. By equivalently formulating the posterior as a composition of local marginals, our approach enables expressive $k$-order dependency modeling. To ensure tractability, we proposed an iterative procedure that guarantees positive definiteness of the resulting correlation matrix via conditional correlation parameterizations. This formulation enables reparameterized sampling and allows efficient optimization. We further generalized the model to support tree-structured backbone dependencies, enabling flexible incorporation of more structured latent correlations. Empirical results across diverse tasks, including time series modeling and constrained clustering, demonstrate the effectiveness of our method in capturing complex dependency structures and improving downstream performance.

**Limitations & Future Work** The proposed method requires specifying a backbone structure to construct higher-order correlations. This limitation is mitigated by generalizing to a learnable tree-structured backbones. For future work, we will investigate the combination of instance-level and dimension-level correlation structure, to further enhance the expressivity of posterior approximation.

**Acknowledgement** This work is supported by the National Natural Science Foundation of China (No. 62276280), Guangzhou Science and Technology Planning Project (No. 2024A04J9967).

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

# A Proofs

Followings are the details of proofs of the claim from the main text - Theorem 2.1, Theorem 2.2, Corollary 2.3 and 2.4.

## A.1 Proof of Theorem 2.1

**Lemma A.1.** *Suppose that $\mathbf{z}_A$ and $\mathbf{z}_B$ are conditionally independent given $\mathbf{z}_C$ where $A, B, C \subseteq \{i, i+1, \cdots, i+k\}$ are mutually exclusive, then in terms of probabilities,*

$$q(\mathbf{z}_A, \mathbf{z}_B, \mathbf{z}_C) = \frac{q(\mathbf{z}_A, \mathbf{z}_C)q(\mathbf{z}_B, \mathbf{z}_C)}{q(\mathbf{z}_C)}. \tag{13}$$

*Proof.* According to the definition of conditional independence,

$$q(\mathbf{z}_A, \mathbf{z}_B | \mathbf{z}_C) = q(\mathbf{z}_A | \mathbf{z}_C)q(\mathbf{z}_B | \mathbf{z}_C) = \frac{q(\mathbf{z}_A, \mathbf{z}_C)q(\mathbf{z}_B, \mathbf{z}_C)}{q(\mathbf{z}_C)^2}. \tag{14}$$

Multiplying both sides by $q(\mathbf{z}_C)$, we can obtain

$$q(\mathbf{z}_A, \mathbf{z}_B, \mathbf{z}_C) = \frac{q(\mathbf{z}_A, \mathbf{z}_C)q(\mathbf{z}_B, \mathbf{z}_C)}{q(\mathbf{z}_C)}. \tag{15}$$

$\square$

**Theorem A.2.** *For any joint distribution $q(\mathbf{z}) = \mathcal{N}(\mathbf{z}; \boldsymbol{\mu}, \mathbf{P}^{-1})$ with a precision matrix $\mathbf{P}$ that has a $k$-order connection structure, it can be equivalently expressed as*

$$q(\mathbf{z}) = \prod_{i=1}^{N-k+1} q(\mathbf{z}_{i:i+k-1}) \prod_{i=1}^{N-k} \frac{q(\mathbf{z}_{i:i+k})}{q(\mathbf{z}_{i:i+k-1})q(\mathbf{z}_{i+1:i+k})}, \tag{16}$$

*where $q(\mathbf{z}_{i:i+k-1})$ and $q(\mathbf{z}_{i:i+k})$ are the marginals of $q(\mathbf{z})$ over $\mathbf{z}_{i:i+k-1}$ and $\mathbf{z}_{i:i+k}$. Moreover, if the $(k+1)$-variate marginals $q(\mathbf{z}_{i:i+k})$ are valid distribution for all $i = 1, 2, \cdots, N-k$, then $q(\mathbf{z})$ will also be a valid distribution.*

*Proof.* To prove the theorem, we turn to prove that for any $t - s \geq k$ and $s \in \{1, \cdots, N-t\}$, the (marginal) distribution of $\mathbf{z}_{s:t}$ is given by

$$q(\mathbf{z}_{s:t}) = \prod_{i=s}^{t-k+1} q(\mathbf{z}_{i:i+k-1}) \prod_{i=s}^{t-k} \frac{q(\mathbf{z}_{i:i+k})}{q(\mathbf{z}_{i:i+k-1})q(\mathbf{z}_{i+1:i+k})}. \tag{17}$$

The result is trivial for $t - s = k$, where the right-hand side becomes

$$q(\mathbf{z}_{s:s+k-1})q(\mathbf{z}_{s+1:s+k})\frac{q(\mathbf{z}_{s:s+k})}{q(\mathbf{z}_{s:s+k-1})q(\mathbf{z}_{s+1:s+k})} = q(\mathbf{z}_{s:s+k}) = q(\mathbf{z}_{s:t}). \tag{18}$$

To start the induction proof, we first prove it for $t - s = k + 1$, which is

$$\begin{aligned}
q(\mathbf{z}_{s:s+k+1}) &= q(\mathbf{z}_{s:s+k-1})q(\mathbf{z}_{s+1:s+k})q(\mathbf{z}_{s+2:s+k+1}) \\
&\quad \times \frac{q(\mathbf{z}_{s:s+k})}{q(\mathbf{z}_{s:s+k-1})q(\mathbf{z}_{s+1:s+k})} \frac{q(\mathbf{z}_{s+1:s+k+1})}{q(\mathbf{z}_{s+1:s+k})q(\mathbf{z}_{s+2:s+k+1})} \\
&= \frac{q(\mathbf{z}_{s:s+k})q(\mathbf{z}_{s+1:s+k+1})}{q(\mathbf{z}_{s+1:s+k})}.
\end{aligned} \tag{19}$$

By letting $A = \{s\}$, $B = \{s + k + 1\}$ and $C = \{s + 1, \cdots, s + k\}$, then the equation above is a direct conclusion of Lemma A.1, where we use the condition that $z_s$ and $z_{s+k+1}$ are conditionally independent given $\mathbf{z}_B = \mathbf{z}_{s+1:s+k}$ implied by the $k$-order dependency structure. We proceed by

induction and go from $t - s \leq l$ to $t - s = l + 1$. The induction hypothesis gives us

$$q(\mathbf{z}_{s:t}) = \prod_{i=s}^{t-k+1} q(\mathbf{z}_{i:i+k-1}) \prod_{i=s}^{t-k} \frac{q(\mathbf{z}_{i:i+k})}{q(\mathbf{z}_{i:i+k-1})q(\mathbf{z}_{i+1:i+k})}, \tag{20}$$

for any $k \leq t - s \leq l$, and we want to show that for $t - s = l + 1$,

$$q(\mathbf{z}_{s:s+l+1}) = \prod_{i=s}^{s+l-k+2} q(\mathbf{z}_{i:i+k-1}) \prod_{i=s}^{s+l-k+1} \frac{q(\mathbf{z}_{i:i+k})}{q(\mathbf{z}_{i:i+k-1})q(\mathbf{z}_{i+1:i+k})}. \tag{21}$$

Letting $A = \{s\}$, $B = \{s+l+1\}$ and $C = \{s+1, \cdots, s+l\}$ and then applying Lemma A.1 gives us

$$q(\mathbf{z}_{s:s+l+1}) = \frac{q(\mathbf{z}_{s:s+l})q(\mathbf{z}_{s+1:s+l+1})}{q(\mathbf{z}_{s+1:s+l})}, \tag{22}$$

where we can use the induction hypothesis to obtain

$$q(\mathbf{z}_{s:s+l}) = \prod_{i=s}^{s+l-k+1} q(\mathbf{z}_{i:i+k-1}) \prod_{i=s}^{s+l-k} \frac{q(\mathbf{z}_{i:i+k})}{q(\mathbf{z}_{i:i+k-1})q(\mathbf{z}_{i+1:i+k})},$$

$$q(\mathbf{z}_{s+1:s+l+1}) = \prod_{i=s+1}^{s+l-k+2} q(\mathbf{z}_{i:i+k-1}) \prod_{i=s+1}^{s+l-k+1} \frac{q(\mathbf{z}_{i:i+k})}{q(\mathbf{z}_{i:i+k-1})q(\mathbf{z}_{i+1:i+k})}, \tag{23}$$

$$q(\mathbf{z}_{s+1:s+l}) = \prod_{i=s+1}^{s+l-k+1} q(\mathbf{z}_{i:i+k-1}) \prod_{i=s+1}^{s+l-k} \frac{q(\mathbf{z}_{i:i+k})}{q(\mathbf{z}_{i:i+k-1})q(\mathbf{z}_{i+1:i+k})}.$$

Leveraging them to simplify the right-hand side of Eq. (22), we can obtain

$$\begin{aligned}
\text{RHS} &= q(\mathbf{z}_{s:s+k-1}) \times \frac{q(\mathbf{z}_{s:s+k})}{q(\mathbf{z}_{s:s+k-1})q(\mathbf{z}_{s+1:s+k})} \\
&\quad \times \prod_{i=s+1}^{s+l-k+2} q(\mathbf{z}_{i:i+k-1}) \prod_{i=s+1}^{s+l-k+1} \frac{q(\mathbf{z}_{i:i+k})}{q(\mathbf{z}_{i:i+k-1})q(\mathbf{z}_{i+1:i+k})} \\
&= \prod_{i=s}^{s+l-k+2} q(\mathbf{z}_{i:i+k-1}) \prod_{i=s}^{s+l-k+1} \frac{q(\mathbf{z}_{i:i+k})}{q(\mathbf{z}_{i:i+k-1})q(\mathbf{z}_{i+1:i+k})},
\end{aligned} \tag{24}$$

which completes the induction. Notably, the conclusion is not restricted to chain-structured backbones, but also applies to tree-structured backbones. The corresponding $k$-variate and $(k+1)$-variate local marginals are defined over $k$-vertex and $(k+1)$-vertex cliques, respectively, and the conclusion is built up by the Hammersley-Clifford Theorem. $\square$

## A.2 Proof of Theorem 2.2

**Lemma A.3.** *Let $a, b, c$ be distinct integers in $\{1, 2, \cdots, k\}$ and let $L$ be a subset of $\{1, 2, \cdots, k\} \backslash \{a, b, c\}$. For a correlation matrix $\mathbf{R} \in \mathbb{R}^{(k+1) \times (k+1)}$, we denote $D(L)$ as the determinant of the sub-matrix $\mathbf{R}[L] \triangleq [\mathbf{R}]_{L \times L}$, then*

$$1 - \gamma_{ab|cL}^2 = \frac{D(\{a, b, c\} \cup L)D(\{c\} \cup L)}{D(\{a, c\} \cup L)D(\{b, c\} \cup L)}. \tag{25}$$

*Proof.* If $a, b, c$ are indices not in $L$, then define

$$\mathbf{R}[a, b, c | L] \triangleq \begin{bmatrix} 1 & \gamma_{ab|L} & \gamma_{ac|L} \\ \gamma_{ab|L} & 1 & \gamma_{bc|L} \\ \gamma_{ac|L} & \gamma_{bc|L} & 1 \end{bmatrix}, \tag{26}$$

and define $\mathbf{R}[a,b|L]$, $\mathbf{R}[a,c|L]$, $\mathbf{R}[b,c|L]$ as principal $2 \times 2$ sub-matrices of $\mathbf{R}[a,b,c|L]$. Since that

$$\gamma_{ab|cL} = \frac{\gamma_{ab|L} - \gamma_{bc|L}\gamma_{bc|L}}{\sqrt{1 - \gamma_{ac|L}^2}\sqrt{1 - \gamma_{bc|L}^2}}, \tag{27}$$

then

$$\begin{aligned}
1 - \gamma_{ab|cL}^2 &= \frac{(1 - \gamma_{ac|L}^2)(1 - \gamma_{bc|L}^2) - (\gamma_{ab|L} - \gamma_{ac|L}\gamma_{bc|L})^2}{(1 - \gamma_{ac|L}^2)(1 - \gamma_{bc|L}^2)} \\
&= \frac{1 - \gamma_{ac|L}^2 - \gamma_{bc|L}^2 - \gamma_{ab|L}^2 + 2\gamma_{ac|L}\gamma_{bc|L}\gamma_{ab|L}}{(1 - \gamma_{ac|L}^2)(1 - \gamma_{bc|L}^2)} \\
&= \frac{\det \mathbf{R}[abc|L]}{\det \mathbf{R}[ac|L]\det \mathbf{R}[bc|L]}.
\end{aligned} \tag{28}$$

If $L = \varnothing$, then the above becomes

$$\frac{\det \mathbf{R}[abc]}{\det \mathbf{R}[ac]\det \mathbf{R}[bc]} = \frac{D(\{a,b,c\})D(\{c\})}{D(\{a,c\})D(\{b,c\})} \tag{29}$$

since by definition $D(\{c\}) = 1$. Otherwise for $L \neq \varnothing$, let $(z_i, z_j, z_k, \mathbf{z}_L)$ be a mean zero normal random vector with correlation matrix $\mathbf{R}[\{a,b,c\} \cup L]$ and unit variances. Let $V_{abc} = \mathrm{diag}(\mathrm{Var}[z_a|\mathbf{z}_L], \mathrm{Var}[z_b|\mathbf{z}_L], \mathrm{Var}[z_c|\mathbf{z}_L])$ so that $V_{abc}^{1/2}\mathbf{R}[abc|L]V_{abc}^{1/2}$ is the covariance matrix of $(z_a, z_b, z_c)|\mathbf{z}_L$. Since the determinant of a positive definite matrix can be decomposed as the multiplication of determinant of its principal sub-matrix and determinant of the corresponding Schur complement, then

$$\det(V_{abc}^{1/2}\mathbf{R}[abc|L]V_{abc}^{1/2}) = \frac{\det \mathbf{R}[\{a,b,c\} \cup L]}{\det \mathbf{R}[L]} = \frac{D(\{a,b,c\} \cup L)}{D(L)}, \tag{30}$$

so that

$$\det \mathbf{R}[abc|L] = \frac{D(\{a,b,c\} \cup L)}{D(L)\,\mathrm{Var}[z_a|\mathbf{z}_L]\,\mathrm{Var}[z_b|\mathbf{z}_L]\,\mathrm{Var}[z_c|\mathbf{z}_L]}. \tag{31}$$

Similarly,

$$\det \mathbf{R}[ac|L]\det \mathbf{R}[bc|L] = \frac{D(\{a,c\} \cup L)D(\{b,c\} \cup L)}{D^2(L)\,\mathrm{Var}[z_a|\mathbf{z}_L]\,\mathrm{Var}[z_b|\mathbf{z}_L]\,\mathrm{Var}[z_c|\mathbf{z}_L]^2}. \tag{32}$$

Hence,

$$\frac{\det \mathbf{R}[abc|L]}{\det \mathbf{R}[ac|L]\det \mathbf{R}[bc|L]} = \frac{D(\{a,b,c\} \cup L)D(L)\,\mathrm{Var}[z_c|\mathbf{z}_L]}{D(\{a,c\} \cup L)D(\{b,c\} \cup L)}. \tag{33}$$

By another application of the determinant decomposition, $D(L)\,\mathrm{Var}[z_c|\mathbf{z}_L] = D(\{c\} \cup L)$, which completes the proof. $\qquad\square$

**Lemma A.4.** *For a correlation matrix* $\mathbf{R} \in \mathbb{R}^{(k+1) \times (k+1)}$ *with conditional correlations* $\gamma_{ij|\mathcal{I}_{ij}}$ *as defined by Eq.* (8) *with* $|i - j| \leq k$, *its determinant is given by*

$$\det \mathbf{R} = \prod_{i=1}^{k}(1 - \gamma_{i,i+1}^2)\prod_{t=2}^{k}\prod_{j=1}^{k+1-t}(1 - \gamma_{j,j+t|\mathcal{I}_{j,j+t}}^2) \tag{34}$$

*Proof.* The result is known for $k = 1$. To start the induction proof, we first prove it for $k = 2$. As a special case of conditional correlation as defined by Eq. (8),

$$\gamma_{13|2} = \frac{\gamma_{13} - \gamma_{12}\gamma_{23}}{\sqrt{1 - \gamma_{12}^2}\sqrt{1 - \gamma_{23}^2}}, \tag{35}$$

so that

$$1 - \gamma_{13|2}^2 = \frac{1 - \gamma_{12}^2 - \gamma_{23}^2 - \gamma_{13}^2 + 2\gamma_{12}\gamma_{23}\gamma_{13}}{(1 - \gamma_{12}^2)(1 - \gamma_{23}^2)} = \frac{\det(\mathbf{R})}{(1 - \gamma_{12}^2)(1 - \gamma_{23}^2)}. \tag{36}$$

Hence $\det(\mathbf{R}) = (1 - \gamma_{12}^2)(1 - \gamma_{23}^2)(1 - \gamma_{13|2}^2)$. We proceed by induction and go from $k$ to $k + 1$. The induction hypothesis gives us

$$\det \mathbf{R}[\{1, \cdots, k\}] = D(\{1, \cdots, k\}) = \prod_{i=1}^{k-1}(1 - \gamma_{i,i+1}^2) \prod_{t=2}^{k-1} \prod_{i=1}^{k-t}(1 - \gamma_{i,i+t|\mathcal{I}_{i,i+t}}^2), \quad (37)$$

and we want to show that $\det \mathbf{R}$ for size $k \times k$ is

$$\prod_{i=1}^{k}(1 - \gamma_{i,i+1}^2) \prod_{t=2}^{k} \prod_{i=1}^{k+1-t}(1 - \gamma_{i,i+t|\mathcal{I}_{i,i+t}}) = D(\{1, \cdots, k\})(1 - \gamma_{k,k+1}^2)$$

$$(1 - \gamma_{k-1,k+1|k}^2) \cdots (1 - \gamma_{1,k+1|\mathcal{I}_{1,k+1}}^2). \quad (38)$$

By Lemma A.3, this is:

$$D(\{1, \cdots, k\})D(\{k, k+1\}) \frac{D(k-1, k, k+1)D(\{k\})}{D(\{k-1, k\})D(\{k, k+1\})}$$

$$\times \frac{D(\{k-2, k-1, k, k+1\})D(\{k-1, k\})}{D(\{k-2, k-1, k\})D(\{k-1, k, k+1\})}$$

$$\times \cdots \times \frac{\det \mathbf{R} D(\{2, \cdots, k\})}{D(\{1, \cdots, k\})D(\{k, k+1\})} \quad (39)$$

$$= D(\{1, \cdots, k\})D(\{k, k+1\}) \prod_{t=2}^{k} \frac{D(k+1-t, \cdots, k+1)D(\{k-t+2, \cdots, k\})}{D(\{k+1-t, \cdots, k\})D(\{k-t+2, \cdots, k+1\})}$$

$$= D(\{1, \cdots, k\})D(\{k, k+1\}) \frac{D(\{k\})}{D(\{1, \cdots, k\})} \frac{D(\{1, \cdots, k+1\})}{D(\{k, k+1\})}$$

$$= \det \mathbf{R} \times D(\{k\}) = \det \mathbf{R}.$$

$$\square$$

**Theorem A.5.** *By writing the correlation matrix* $\mathbf{R}^{(i)}$ *as the following partitioned form*

$$\mathbf{R}^{(i)} = \begin{bmatrix} 1 & \gamma_{i,i+1} & \cdots & \gamma_{i,i+k-1} & \gamma_{i,i+k} \\ \gamma_{i,i+1} & 1 & \cdots & \gamma_{i+1,i+k-1} & \gamma_{i+1,i+k} \\ \vdots & \vdots & \ddots & \vdots & \vdots \\ \gamma_{i,i+k-1} & \gamma_{i+1,i+k-1} & \cdots & 1 & \gamma_{i+k-1,i+k} \\ \gamma_{i,i+k} & \gamma_{i+1,i+k} & \cdots & \gamma_{i+k-1,i+k} & 1 \end{bmatrix}, \quad (40)$$

*if the upper-left sub-matrix* $[\mathbf{R}^{(i)}]_{1:k,1:k}$ *and the lower-right sub-matrix* $[\mathbf{R}^{(i)}]_{2:k+1,2:k+1}$ *in the dotted frames are both positive definite, also if* $|\gamma_{i,i+k}^c| < 1$ *and we set* $\gamma_{i,i+k} = \mathcal{M}(\gamma_{i,i+k}^c)$, *then* $\mathbf{R}^{(i)}$ *will be positive definite.*

*Proof.* Without loss of generality, we use the indices $\{1, 2, \cdots, k+1\}$ to replace the original indices $\{i, i+1, \cdots, i+k\}$ of the correlation $\mathbf{R}^{(i)} \in \mathbb{R}^{(k+1) \times (k+1)}$, giving the correlation matrix

$$\mathbf{R} = \begin{bmatrix} 1 & \gamma_{12} & \cdots & \gamma_{1k} & \gamma_{1,k+1} \\ \gamma_{12} & 1 & \cdots & \gamma_{2k} & \gamma_{2,k+1} \\ \vdots & \vdots & \ddots & \vdots & \vdots \\ \gamma_{1k} & \gamma_{2k} & \cdots & 1 & \gamma_{k,k+1} \\ \gamma_{1,k+1} & \gamma_{2,k+1} & \cdots & \gamma_{k,k+1} & 1 \end{bmatrix} \quad (41)$$

To prove that $\mathbf{R}$ is positive definite, we only need to show that $\det \mathbf{R} > 0$, given the sub-matrix $[\mathbf{R}]_{1:k,1:k}$ is positive definite. Since that both the sub-matrices $[\mathbf{R}]_{1:k,1:k}$ and $[\mathbf{R}]_{2:k+1,2:k+1}$ are positive definite, then the corresponding marginal distributions $q(\mathbf{z}_{1:k})$ and $q(\mathbf{z}_{2:k+1})$ must be valid, so we can define conditional correlations $\gamma_{ij|\mathcal{I}_{ij}}$ for any $|i - j| < k$, satisfying that $|\gamma_{ij|\mathcal{I}_{ij}}| < 1$. Combing the provided $(k+1)$-th order conditional $\gamma_{i,i+k|\mathcal{I}_{i,i+k}}$, we can leverage Lemma A.4 to

compute the determinant of the correlation matrix $\mathbf{R}$ as follows

$$\det \mathbf{R} = \prod_{i=1}^{k}(1 - \gamma_{i,i+1}^2)\prod_{t=2}^{k}\prod_{j=1}^{k+1-t}(1 - \gamma_{j,j+t|\mathcal{I}_{j,j+t}}^2), \tag{42}$$

which is guaranteed to be positive, given that $\gamma_{i,i+k|\mathcal{I}_{i,i+k}}$ also lies in $(-1, 1)$. Therefore, $\det \mathbf{R} > 0$ and then $\mathbf{R}$ is positive definite. $\qquad\square$

## A.3 Proof of Corollary 2.3 and 2.4

**Corollary A.6.** *If all correlation coefficients in $\mathbf{\Gamma}_1 = \{\gamma_{i,i+1}\}_{i=1}^{N-1}$ and $\mathbf{\Gamma}_t = \{\gamma_{i,i+t}^c\}_{i=1}^{N-t}$ for $t = 2, 3, \cdots, k$ lie in the interval $(-1, 1)$, then we can use them to construct a $(k + 1) \times (k + 1)$ correlation matrix $\mathbf{R}^{(i)}$ with k-order dependency.*

*Proof.* The result is known for $k = 1$, since the determinant $\det \mathbf{R}^{(i)} = 1 - \gamma_{i,i+1}^2 > 0$. To start the induction proof, we first prove it for $k = 2$, which is to show that

$$\mathbf{R}^{(i)} = \begin{bmatrix} 1 & \gamma_{i,i+1} & \gamma_{i,i+2} \\ \gamma_{i,i+1} & 1 & \gamma_{i+1,i+2} \\ \gamma_{i,i+2} & \gamma_{i+1,i+2} & 1 \end{bmatrix} \tag{43}$$

is positive definite with $\gamma_{i,i+2} = \mathcal{M}(\gamma_{i,i+2}^c)$ and $|\gamma_{i,i+2}| < 1$. In this case, the $2 \times 2$ upper-left submatrix $[\mathbf{R}^{(i)}]_{1:2,1:2}$ of $\mathbf{R}^{(i)}$ is positive definite, since its determinant

$$\det \begin{vmatrix} 1 & \gamma_{i,i+1} \\ \gamma_{i,i+1} & 1 \end{vmatrix} = 1 - \gamma_{i,i+1}^2 > 0, \tag{44}$$

for $|\gamma_{i,i+1}| < 1$. And similarly, the lower-right submatrix $[\mathbf{R}^{(i)}]_{2:3,2:3}$ is also positive definite. Leveraging the conclusion of Theorem 2.2 and the condition that $\gamma_{i,i+2} = \mathcal{M}(\gamma_{i,i+2}^c)$ with $|\gamma_{i,i+2}^c| < 1$, we can guarantee $\mathbf{R}^{(i)}$ to be positive definite. We proceed by induction and go from $k - 1$ to $k$, where we want to show that the correlation matrix

$$\mathbf{R}^{(i)} = \begin{bmatrix} 1 & \gamma_{i,i+1} & \cdots & \gamma_{i,i+k-1} & \gamma_{i,i+k} \\ \gamma_{i,i+1} & 1 & \cdots & \gamma_{i+1,i+k-1} & \gamma_{i+1,i+k} \\ \vdots & \vdots & \ddots & \vdots & \vdots \\ \gamma_{i,i+k-1} & \gamma_{i+1,i+k-1} & \cdots & 1 & \gamma_{i+k-1,i+k} \\ \gamma_{i,i+k} & \gamma_{i+1,i+k} & \cdots & \gamma_{i+k-1,i+k} & 1 \end{bmatrix}, \tag{45}$$

is positive definite. The induction hypothesis gives us the upper-left submatrix $[\mathbf{R}^{(i)}]_{1:k,1:k}$ and lower-right submatrix $[\mathbf{R}^{(i)}]_{2:k+1,2:k+1}$ are both positive definite, by ensuring that correlation coefficients $\mathbf{\Gamma}^1$ and $\mathbf{\Gamma}^t$ for $t \in \{2, \cdots, k-1\}$ lie in the interval $(-1, 1)$. By further ensuring that $\gamma_{i,i+k} = \mathcal{M}(\gamma_{i,i+k}^c)$ with $|\gamma_{i,i+k}^c| < 1$, then we can leverage Theorem 2.2 to guarantee the positive definiteness of $\mathbf{R}^{(i)}$, which completes the proof. $\qquad\square$

**Corollary A.7.** *If the first-order correlations $\mathbf{\Gamma}_1$ and higher-order conditional correlations $\mathbf{\Gamma}_t$ for $t = 2, 3, \cdots, k$ are built upon a tree-structured backbone, and all correlation parameters lie in the interval $(-1, 1)$, then we can use them to construct a $(k + 1) \times (k + 1)$ correlation matrix $\mathbf{R}^{(i)}$ with k-order dependency structure.*

*Proof.* This corollary of the tree-structured backbone can be similarly proved as above by induction. Notice that every $(k + 1)$-vertex clique $\mathbb{C} \in \mathcal{C}_{k+1}$ can be decomposed into two $k$-vertex cliques $\mathbb{C}_1$ and $\mathbb{C}_2$ such that $\mathbb{C} = \mathbb{C}_1 \cup \mathbb{C}_2$ and $|\mathbb{C}_1 \cap \mathbb{C}_2| = k - 1$. So the correlation matrix with respect to $\mathbb{C}$ can be partitioned as two submatrices corresponding to $\mathbb{C}_1$ and $\mathbb{C}_2$, respectively. By ensuring their positive definiteness and letting the $k$-order conditional correlation lie in the interval $(-1, 1)$, we can similarly guarantee the correlation matrix of $\mathbb{C}$ to be positive definite. Therefore, we can perform induction by starting from $k = 2$, and sequentially expand the correlation matrix by introducing higher-order conditional correlations and ensuring them to lie in the interval $(-1, 1)$. As the induction

completes, we can use these correlation coefficients to construct a $(k+1) \times (k+1)$ correlation matrix $\mathbf{R}^{(i)}$. $\qquad\qquad\qquad\qquad\qquad\qquad\qquad\qquad\qquad\qquad\qquad\qquad\qquad$ $\square$

## B  Procedure of Constructing the Correlation Matrix

---

**Algorithm 1** Algorithm of constructing the correlation matrix $\mathbf{R}$

---

**Input:** Conditional parameters $\boldsymbol{\Gamma}_1 = \{\gamma_{i,i+1}\}_{i=1}^{N-1}, \boldsymbol{\Gamma}_2 = \{\gamma_{i,i+2}^c\}_{i=1}^{N-2}, \cdots, \boldsymbol{\Gamma}_K = \{\gamma_{i,i+K}^c\}_{i=1}^{N-K}$
**Output:** Full correlation matrix $\mathbf{R}$ of size $N \times N$

 1: **function** CORRELATION_MATRIX_CONSTRUCTION()
 2: $\qquad$ $\mathbf{R} \leftarrow \mathbf{I}_N$ $\qquad\qquad\qquad\qquad\qquad\qquad\qquad\qquad\qquad\qquad\qquad\qquad$ ▷ Identity matrix
 3: $\qquad$ $k \leftarrow 1$ $\qquad\qquad\qquad\qquad\qquad\qquad\qquad\qquad\qquad\qquad\qquad$ ▷ Starting from the first-order
 4: $\qquad$ **for** $i \leftarrow 1$ to $N$ **do**
 5: $\qquad\qquad$ $\mathbf{R}[i, i+1] \leftarrow \gamma_{i,i+1}$
 6: $\qquad\qquad$ $\mathbf{R}[i+1, i] \leftarrow \gamma_{i,i+1}$
 7: $\qquad$ **end for**
 8: $\qquad$ **for** $k \leftarrow 2$ to $K$ **do** $\qquad\qquad\qquad\qquad\qquad\qquad\qquad\qquad$ ▷ Loop through higher orders
 9: $\qquad\qquad$ **for** $i \leftarrow 1$ to $N-k$ **do**
10: $\qquad\qquad\qquad$ $\gamma_{i,i+k} \leftarrow$ inverse_conditional$(\gamma_{i,i+k}^c, \mathbf{R}[i:i+k, i:i+k])$ $\qquad$ ▷ Inverting Eq. (8)
11: $\qquad\qquad\qquad$ $\mathbf{R}[i, i+k] \leftarrow \gamma_{i,i+k}$
12: $\qquad\qquad\qquad$ $\mathbf{R}[i+k, i] \leftarrow \gamma_{i,i+k}$
13: $\qquad\qquad$ **end for**
14: $\qquad$ **end for**
15: $\qquad$ **return** $\mathbf{R}$
16: **end function**

---

## C  Evidence Lower Bound

The evidence lower bound of our proposed method is given by

$$\mathcal{L}(\boldsymbol{\theta}, \boldsymbol{\phi}, \mathbf{x}) = \log p_{\boldsymbol{\theta}}(\mathbf{x}, \tilde{\mathbf{z}}) + \mathcal{H}[q_{\boldsymbol{\phi}}(\mathbf{z})], \tag{46}$$

where $\tilde{\mathbf{z}}$ denotes the re-parameterized latent variables. The first term above can be directly computed by

$$\log p_{\boldsymbol{\theta}}(\mathbf{x}, \tilde{\mathbf{z}}) = \sum_{i=1}^{N} \log p_{\boldsymbol{\theta}}(\mathbf{x}_i | \tilde{\mathbf{z}}_i) + \log p(\tilde{z}_i), \tag{47}$$

where $\tilde{z}_i$ is the reparameterization for latent variable $z_i$, $i = 1, \cdots, N$. And the entropy of the posterior

$$q_{\boldsymbol{\phi}}(\mathbf{z}) = \prod_{i=1}^{N-k+1} q_{\boldsymbol{\phi}}(\mathbf{z}_{i:i+k-1}) \prod_{i=1}^{N-k} \frac{q_{\boldsymbol{\phi}}(\mathbf{z}_{i:i+k})}{q_{\boldsymbol{\phi}}(\mathbf{z}_{i:i+k-1}) q(\mathbf{z}_{i+1:i+k})} \tag{48}$$

with $k$-order dependency structure can be factorized as entropy terms with respect to $k$-variate and $(k+1)$-variate local marginals

$$\mathcal{H}[q_{\boldsymbol{\phi}}(\mathbf{z})] = \sum_{i=1}^{N-k+1} \mathcal{H}[q_{\boldsymbol{\phi}}(\mathbf{z}_{i:i+k-1})] + \sum_{i=1}^{N-k} \mathcal{H}[q_{\boldsymbol{\phi}}(\mathbf{z}_{i:i+k})] - \mathcal{H}[q_{\boldsymbol{\phi}}(\mathbf{z}_{i:i+k-1})] - \mathcal{H}[q(\mathbf{z}_{i+1:i+k})]$$

$$= \sum_{i=1}^{N-k} \mathcal{H}[q_{\boldsymbol{\phi}}(\mathbf{z}_{i:i+k})] - \sum_{i=2}^{N-k} \mathcal{H}[q_{\boldsymbol{\phi}}(\mathbf{z}_{i:i+k-1})], \tag{49}$$

where the entropy of each normally distributed local marginal can be directly computed by its mean and covariance.

## D Experimental Details

### D.1 Datasets

The datasets used in the time series anomaly detection task are the followings:

- **SMAP (Soil Moisture Active Passive):** NASA's Soil Moisture Active Passive mission [25] aims to measure global soil moisture and freeze/thaw states to enhance understanding of Earth's water, energy, and carbon cycles. The SMAP dataset comprises multivariate time series telemetry data collected from the SMAP satellite, including a training and a testing subsets. It includes expert-labeled anomalies in testing subsets, making it suitable for benchmarking time series anomaly detection algorithms.

- **MSL (Mars Science Laboratory):** Originates from NASA's Mars Science Laboratory mission [25], featuring the Curiosity rover, explores Mars' surface to assess its habitability. The MSL dataset contains multivariate time series telemetry data from the Curiosity rover, with expert annotations identifying anomalous events in the testing subsets.

- **SMD (Server Machine Dataset):** Collected by researchers from a large Internet company [55]. SMD comprises a 5-week-long collection of multivariate time series data from 28 server machines, each monitored by 38 sensors capturing metrics like CPU usage, memory, and network throughput. The dataset includes labeled anomalies, facilitating supervised learning approaches. Due to the high degree of similarity in temporal characteristics across servers, we conducted experiments solely on machine 1-1 for simplicity.

The datasets used in the time series forecasting task are the followings:

- **ETT (Electricity Transformer Temperature):** This dataset includes the target variable "oil temperature" along with six power load features [69]. It is recorded at two different frequencies: hourly (*i.e.*, ETTh1 and ETTh2) and every 15 minutes (*i.e.*, ETTm1 and ETTm2), spanning a period of two years.

- **Electricity:** Sourced from the UCI Machine Learning Repository[3] and preprocessed following [33], this dataset contains hourly electricity consumption (in kWh) for 321 clients from 2012 to 2014.

- **Exchange:** This dataset comprises daily exchange rates for eight countries, collected from 1990 to 2016 [44].

- **Weather[4]:** Includes 21 meteorological indicators (e.g., temperature, humidity), recorded every 10 minutes throughout the year 2020.

The datasets utilized in the constrained clustering task are as follows:

- **MNIST:** A widely used benchmark dataset containing 70,000 grayscale images of handwritten digits. Each image is represented as a 784-dimensional vector by flattening the original 28×28 pixel grid [35].

- **Fashion MNIST:** A collection of Zalando's fashion article images [62], this dataset includes a training set of 60,000 images and a test set of 10,000 images.

- **Reuters:** Contains 810,000 English news articles [36]. Following the preprocessing method of DEC [64], we select four root categories—corporate/industrial, government/social, markets, and economics—and exclude documents with multiple labels. The resulting dataset contains 685,071 articles, each represented using tf-idf features over the top 2,000 words. A random subset of 10,000 documents is used for experiments.

- **STL-10:** Composed of 96×96 color images across 10 object classes, with 13,000 labeled samples [16]. For feature extraction, we apply a ResNet-50 model as done in VaDE [27].

---

[3]https://archive.ics.uci.edu/ml/datasets/ElectricityLoadDiagrams
[4]https://www.bgc-jena.mpg.de/wetter

Table 5: Detailed information of datasets used in time series anomaly detection and forecasting tasks.

| Tasks | Dataset | Dim | Size (Train, Validation, Test) | Domain |
|---|---|---|---|---|
| Forecasting | ETTm1 | 7 | (34465, 11521, 11521) | Electricity |
| | ETTh1 | 7 | (8545, 2881, 2881) | Electricity |
| | Electricity | 321 | (18317, 2633, 5261) | Electricity |
| | Weather | 21 | (36792, 5271, 10540) | Weather |
| | Exchange | 8 | (5120, 665, 1422) | Exchange rate |
| Anomaly Detection | SMD | 38 | (566724, 141681, 708420) | Server Machine |
| | MSL | 55 | (44653, 11664, 73729) | Spacecraft |
| | SMAP | 25 | (108146, 27037, 427617) | Spacecraft |

## D.2 Further Experiments

We also run our model under univariate forecasting settings, where only a single feature is considered in each dataset. The experimental results in Table 6 shows that our method outperforms other fundamental time series modeling techniques. The superior capability of our method in capturing temporal dependencies is more pronounced in this setting, as all models are restricted to fully exploiting temporal correlations without leveraging inter-channel information.

Table 6: Univariate time series forecasting comparisons. Best performance is highlighted in bold font and the second best results are underlined.

| Method | | VRAE | | Informer | | Autoformer | | TCN | | Ours | | | | | |
|---|---|---|---|---|---|---|---|---|---|---|---|---|---|---|---|---|
| | | | | | | | | | | | 1-order | | 3-order | | 10-order | |
| | | MSE | MAE | MSE | MAE | MSE | MAE | MSE | MAE | MSE | MAE | MSE | MAE | MSE | MAE |
| ETTh1 | 24 | 0.059 | 0.215 | 0.098 | 0.247 | 0.057 | 0.189 | 0.104 | 0.254 | 0.054 | 0.178 | 0.038 | 0.149 | **0.032** | **0.127** |
| | 48 | 0.097 | 0.279 | 0.158 | 0.319 | 0.070 | 0.207 | 0.206 | 0.366 | 0.087 | 0.229 | 0.061 | 0.187 | **0.052** | **0.163** |
| | 168 | 0.191 | 0.402 | 0.183 | 0.346 | 0.108 | 0.260 | 0.462 | 0.586 | 0.161 | 0.316 | 0.131 | 0.278 | **0.088** | **0.212** |
| | 336 | 0.187 | 0.400 | 0.222 | 0.387 | 0.119 | 0.281 | 0.422 | 0.564 | 0.170 | 0.333 | 0.149 | 0.303 | **0.105** | **0.240** |
| | 720 | 0.244 | 0.471 | 0.269 | 0.435 | **0.109** | **0.264** | 0.438 | 0.578 | 0.221 | 0.392 | 0.172 | 0.336 | 0.139 | 0.285 |
| ETTm1 | 24 | 0.021 | 0.122 | 0.030 | 0.137 | 0.022 | 0.115 | 0.027 | 0.127 | 0.018 | 0.101 | 0.015 | 0.091 | **0.012** | **0.075** |
| | 48 | 0.039 | 0.172 | 0.069 | 0.203 | 0.032 | 0.138 | 0.040 | 0.154 | 0.041 | 0.154 | 0.027 | 0.123 | **0.022** | **0.105** |
| | 168 | 0.060 | 0.217 | 0.194 | 0.372 | 0.045 | 0.158 | 0.097 | 0.246 | 0.052 | 0.173 | 0.043 | 0.158 | **0.034** | **0.129** |
| | 336 | 0.143 | 0.344 | 0.401 | 0.554 | **0.071** | 0.207 | 0.305 | 0.455 | 0.131 | 0.276 | 0.091 | 0.229 | 0.073 | **0.192** |
| | 720 | 0.211 | 0.428 | 0.512 | 0.644 | 0.102 | 0.254 | 0.445 | 0.576 | 0.134 | 0.287 | 0.135 | 0.282 | **0.099** | **0.227** |
| Electricity | 24 | 0.370 | 0.459 | 0.251 | 0.275 | 0.290 | 0.411 | 0.243 | 0.367 | 0.252 | 0.278 | 0.247 | 0.285 | **0.166** | **0.249** |
| | 48 | 0.459 | 0.519 | 0.346 | 0.339 | 0.310 | 0.408 | 0.283 | 0.397 | 0.301 | 0.309 | 0.298 | 0.318 | **0.202** | **0.277** |
| | 168 | 0.547 | 0.575 | 0.544 | 0.424 | 0.435 | 0.490 | 0.357 | 0.449 | 0.413 | 0.384 | 0.408 | 0.386 | **0.270** | **0.323** |
| | 336 | 0.682 | 0.660 | 0.713 | 0.512 | 0.646 | 0.606 | 0.355 | 0.446 | 0.551 | 0.468 | 0.537 | 0.468 | **0.339** | **0.369** |
| | 720 | 0.889 | 0.790 | 1.182 | 0.806 | 0.609 | 0.587 | **0.387** | 0.477 | 0.862 | 0.650 | 0.812 | 0.628 | 0.454 | **0.448** |
| Average | | 0.280 | 0.404 | 0.345 | 0.400 | 0.202 | 0.306 | 0.278 | 0.403 | 0.230 | 0.302 | 0.211 | 0.281 | **0.139** | **0.228** |

## D.3 Implementation Details

**Time Series Anomaly Detection**   We set the input sequence length to 100 and use GRU and dense layers with 500 hidden units each. The latent dimension is fixed at 3. Models are trained with a batch size of 50 for up to 20 epochs using early stopping. Optimization is performed using the Adam optimizer with an initial learning rate of $10^{-3}$. L2 regularization with a coefficient of $10^{-4}$ is applied to all layers. During training, 30% of the data is reserved for validation.

**Time Series Forecasting**   We adopt a single-layer fully connected network as the feedforward predictor. The latent representation dimension is set to 128. The model is trained using the Adam optimizer with an initial learning rate of $10^{-3}$, decayed by a factor of 0.95 after each epoch. Early stopping is applied within 10 epochs to prevent overfitting.

**Constrained Clustering.**   To ensure a fair comparison with baseline methods, we adopt the same encoder-decoder feed-forward architecture: four fully connected layers with sizes 500, 500, 2000,

Table 7: Hyperparameters setting of constrained clustering task.

|  | MNIST | fMNIST | Reuters | STL-10 |
|---|---|---|---|---|
| Batch size | 256 | 256 | 256 | 256 |
| Epochs | 1000 | 500 | 500 | 500 |
| Learning rate | 0.001 | 0.001 | 0.001 | 0.001 |
| Decay | 0.9 | 0.9 | 0.9 | 0.9 |
| Epochs decay | 20 | 20 | 20 | 20 |

and $D$ units, respectively, where $D = 10$ unless otherwise specified. For all VAE-based baselines and our proposed methods built on VAE backbones, we apply 10 epochs of pretraining. For DEC-based baselines, we follow their standard training procedure, including 50 epochs of layer-wise pretraining and 100 epochs of fine-tuning. Each dataset is split into training and test sets; model training is conducted on the training split, while all reported results are evaluated on the test split. Pairwise constraints are randomly generated within the training set: a must-link is assigned if two sampled instances share the same label, and a cannot-link otherwise. To ensure consistent training conditions across methods, we uniformly set the absolute constraint strength $|[\mathbf{A}]_{ij}| = 10^4$ and sample 6000 such constraints for all datasets. Following DC-GMM, we use the same set of hyperparameters across all four datasets, detailed in Table 7. All models are trained with an initial learning rate of 0.001, which decays by a factor of 0.9 every 20 epochs.

### D.4 Resource Usage

Experiments were conducted on an internal computing cluster. Each experiment configuration used one NVIDIA GPU (either a 2080TI or 3090TI), 16 CPUs and a total of 24GB of memory.

