# OpenReview forum: "HoT-VI: Reparameterizable Variational Inference for Capturing Instance-Level High-Order Correlations"
_NeurIPS.cc/2025/Conference — NeurIPS 2025 poster_

### Official Review · Reviewer_Evn6 · 2025-07-01

**Clarity:** 3
**Significance:** 3
**Originality:** 3
**Rating:** 5
**Confidence:** 3

**Summary:**

This paper addresses the problem of expanding mean-field Variational Inference (VI) to account for instance-level dependencies in a computationally efficient way. By considering a particular form of Gaussian variational family, the authors are able to incorporate instance-level dependency within a predefined and tunable range into the model. Guidelines are provided to confine the iterates within a suitable region of the optimization space. Experiments on real-world data support the method's practical applicability and efficacy.

**Questions:**

* TreeVI appears to be the most similar work to yours. Could you elaborate on the key differences between your method and TreeVI, and highlight the improvements brought by your approach?
* How does your method compare to concurrent methods like TreeVI in terms of effective runtime?
* While some information about the datasets used in the experiments is provided in Appendix D.1, it is not said exactly how large these datasets actually are and how many observations they contain. For instance, SMAP is a collection of data products. Could you provide more details on this matter?
* Lines 80-81: “the method can easily be extended to take the dimensional-level correlation into account”. Can you expand on this?

**Ethical Concerns:**

["NO or VERY MINOR ethics concerns only"]

**Final Justification:**

The paper introduces a novel, interesting, and empirically well-justified method.
I was originally concerned that the method may be a simple extension of the already-existing TreeVI, but the authors have clarified important differences between both procedures. Responses to other reviewers go into more details about this matter, and the authors are commited to spending more time explaining the key differences in their revised manuscript.
I was also not convinced that the performance gains were worth the added computational complexity. In their rebuttal, the authors provided a runtime study that successfully addressed this concern.
While the method introduced in this paper is strictly limited to the Gaussian setting, I believe that it still appears as a relevant contribution. It is noteworthy that this method achieves state-of-the-art results on a variety of tasks.

**Limitations:**

Yes.

**Paper Formatting Concerns:**

None.

**Quality:**

3

**Strengths And Weaknesses:**

**Strengths**
* The proposed method is original and its implementation seems to be quite simple. There is a clear way to dial between the expresiveness of the variational family and the model's complexity (however this trade-off is not explored in details, see Weaknesses below).
* The method empirically outperforms the baselines in a wide variety of real-world scenarios, including anomaly detection (3 datasets), time series forecasting (4 datasets) and clustering (4 datasets).

**Weaknesses**
* There is no runtime study, Appendix D.4 is not very informative about the computational cost of the method. The trade-off between the gains yielded by more complex models and the added computational cost is not discussed. This is essential for establishing the method's superiority over existing approaches, and should be addressed by the authors.
* The variational posteriors must be Gaussian, hence the method is not applicable to settings where such approximating families are irrelevant, e.g., directional statistics.
* In the constrained clustering experiment, it is unclear how the constraints are incorporated into the model (even with the additional details in Appendix D.3).
* The authors do not discuss how the proposed reparameterization effectively restricts the expressiveness of the variational family.

**Additional comments**
* Line 77: there are two product signs in the equation.
* Table 5: typo in “Epchs”
* Throughout the text: replace "validness" with validity, and "indexes" with indices.

---

> ### Author Rebuttal · Authors · 2025-07-31
>
> We thank the reviewer for the comments and suggestions. Below we respond to each point raised by the reviewer.
>
> **Q1: What is the runtime study for the proposed method, and what is the trade-off between the gains from more complex models and their added computational cost?**
>
> A1: An empirical runtime study was conducted to measure the cost of increasing the approximation order k. Tables 1-3 below show the average training runtime per epoch on different tasks, demonstrating that the computational cost scales linearly with k.
>
> ---
>
> Table 1: Time series anomaly detection performance on SMAP dataset.
> | Methods  | Mean-field | k=1 | k=3 | k=10 | k=50 | k=100 |
> | - | - | - | - | - | - | - |
> | Runtime (s/epoch) | 1.00 | 2.44 | 8.51 | 27.60 | 142.58 | 294.53 |
> | F1     | 0.7774 | 0.8411 | 0.8552 | 0.8636 | 0.8711 | 0.8755 |
> | ELBO   | -109.2182 | -97.6057 | -95.2314 | -92.2948 | -90.4291 | -89.9577 |
>
> ---
>
> Table 2: Time series forecasting performance on ETTh1 dataset with horizon 24.
> | Methods  | Mean-field | k=1 | k=3 | k=10 | k=50 | k=100 |
> | - | - | - | - | - | - | - |
> | Runtime (s/epoch) | 4.80 | 11.45 | 36.94 | 126.44 | 675.75 | 1340.22 |
> | MSE | 0.739 | 0.664 | 0.543 | 0.363 | 0.348 | 0.333 |
> | MAE | 0.716 | 0.570 | 0.505 | 0.376 | 0.362 | 0.352 |
>
> ---
>
> Table 3: Constrained clustering performance on MNIST dataset.
> | Methods  | Mean-field | TreeVI | k=1 | k=3 | k=5 | k=10 | k=50 | k=100 |
> | - | - | - | - | - | - | - | - | - |
> | Runtime (s/epoch) | 0.25 | 0.52 | 0.59 | 1.84 | 3.09 | 6.22 | 29.53 | 60.59 |
> | ACC (\%)     | 96.50 | 97.43 | 97.55 | 98.12 | 98.39 | 98.52 | 98.62 | 98.69 |
> | NMI (\%)     | 91.37 | 93.14 | 93.44 | 93.80 | 94.17 | 94.55 | 94.63 | 94.85 |
> | ARI (\%)     | 92.54 | 93.73 | 93.89 | 94.89 | 95.24 | 95.65 | 95.85 | 96.09 |
>
> ---
>
> The trade-off is between this linear increase in runtime and the corresponding performance gains. While model performance consistently improves as k increases, our experiments show diminishing marginal returns, with higher values of k (such as k beyond 10) yielding progressively smaller gains. Therefore, we focused on k up to 10 in the main paper because this range demonstrates the advantages of our approach while maintaining a practical and efficient balance between performance and computational tractability.
>
> **Q2: What are the specific limitations of the method given that its variational posteriors must be Gaussian, particularly concerning its applicability to settings like directional statistics?**
>
> A2: The Gaussian posterior is a foundational and highly effective choice, used with great success in many state-of-the-art generative models like VAEs and modern diffusion models. We acknowledge that, while powerful, the Gaussian posterior is not universally applicable and is not directly suited for modeling certain data types, such as discrete data or directional statistics. The core contribution of our work is to introduce a novel and scalable method for structuring correlations, and extending this framework to more flexible distributional families to address these broader settings is a promising direction for our future research.
>
> **Q3: How are the constraints specifically incorporated into the model within the constrained clustering experiment?**
>
> A3: As detailed in Section 4.2, the constraints are not implemented as hard rules but are integrated as a probabilistic prior over the cluster assignments $\mathbf{c}$. This soft constraint mechanism guides the model to favor clusterings that are consistent with the provided must-link and cannot-link information. Specifically, the key is to define a conditional probability distribution for the cluster assignments $\mathbf{c}$, conditioned on a pairwise prior information matrix $\mathbf{A}$. This distribution $p( \mathbf{c}|\mathbf{A})$, as shown in Line 345, replaces the uninformative prior $p(\mathbf{c})$ used in standard generative clustering models, to assign a higher probability to clusterings that respect the constraints encoded in $\mathbf{A}$ (If there is a must-link between instances $x_i$ and $x_j$, then $[\mathbf{A}]_ {ij}$ is a  large positive value; if there is a cannot-link between $x_i$ and $x_j$, then $[\mathbf{A}]_ {ij}$ is a large negative value; if there is no constraint, then $[\mathbf{A}]_ {ij}=0$). This prior is then integrated into the overall generative model $p_{\theta}(\mathbf{X}, \mathbf{Z}, \mathbf{c} | \mathbf{A}) = p_{\theta}(\mathbf{X}|\mathbf{Z})p(\mathbf{Z}|\mathbf{c})p(\mathbf{c}|\mathbf{A})$, guiding the learning process within the variational inference framework towards a posterior that respects the given domain knowledge.
>
> **Q4: How does the proposed reparameterization method restrict the expressiveness of the variational family?**
>
> A4: Our reparameterization method trades the universal expressiveness of a general multivariate Gaussian for guaranteed validity and computational tractability. It restricts the variational family to a subset of Gaussian distributions whose precision matrices have a k-order connection structure. And the hyperparameter k allows us to explicitly control the trade-off between expressiveness and computational cost.
>
> **Q5: What are the key differences between the proposed method and TreeVI, what specific improvements does your approach offer, and what is the comparison in terms of effective runtime?**
>
> A5: The key difference between TreeVI and our method is that TreeVI is inherently limited to modeling only first-order correlations, while our method can capture higher-order correlations. This limitation of TreeVI arises from its reliance on an acyclic tree structure to construct its correlation matrix. While this is sufficient for simple pairwise relationships, attempting to model higher-order correlations inherently introduces loops into the underlying correlation structure. The construction that TreeVI uses to build a valid correlation matrix depends on the acyclic property of its backbone and is no longer valid when these loops are present. Moreover, simply modeling higher-order correlation coefficients within the framework of TreeVI does not guarantee the resulting correlation matrix to remain positive definite. So even though TreeVI can capture instance-level correlations, it cannot be easily extended to model higher-order correlations.
>
> Our approach, in contrast, is generalized to explicitly model k-order correlations. For $k=1$, our model is conceptually equivalent to TreeVI, but for $k>1$, it becomes substantially more expressive. As shown in our constrained clustering results (refer to Table 3 above), the runtime of our method is approximately k times that of TreeVI. But even with a small k (e.g., k=3), our method achieves significantly better performance across all metrics compared to TreeVI, demonstrating the benefit of modeling higher-order correlations.
>
> **Q6: More detailed information about the actual sizes of the datasets used in the experiments (e.g., the number of observations for SMAP)?**
>
> A6: The actual sizes of the datasets used in the constrained clustering task can be found in Appendix D.1. Detailed information about the other datasets is shown as follows and will be included in the appendix of our final version.
>
> | Tasks | Dataset | Dim | Size (Train, Validation, Test) | Domain  |
> | :--- | :--- | :-- | :--- | :--- |
> | **Forecasting** | ETTm1 | 7 |  (34465, 11521, 11521) | Electricity |
> | | ETTh1 | 7 |  (8545, 2881, 2881) | Electricity |
> | | Electricity | 321 |  (18317, 2633, 5261) | Electricity |
> | | Weather | 21 | (36792, 5271, 10540) | Weather |
> | | Exchange | 8 |  (5120, 665, 1422) | Exchange rate |
> | **Anomaly Detection** | SMD | 38 |(566724, 141681, 708420) | Server Machine |
> | | MSL | 55 |  (44653, 11664, 73729) | Spacecraft |
> | | SMAP | 25 | (108146, 27037, 427617) | Spacecraft |
>
> **Q7: Clarify extending the method to dimension-level correlations.**
>
> A7: While we focused on instance-level correlations in the paper for clarity, the method can be extended to model dimension-level correlations by introducing a structured covariance to the sampling procedure's noise term.
>
> In the current framework, each dimension is sampled independently, meaning the noise vector $\epsilon$ in the re-parameterization step is drawn from a standard Gaussian with a diagonal covariance matrix $\mathcal{N}(0, I)$. To incorporate dimension-level correlations, we can replace this with a low-rank Gaussian distribution for the noise. By sampling $\epsilon$ from a distribution with a low-rank covariance, we can efficiently induce correlations across the dimensions of the generated latent variables $\mathbf{z}$. This provides a practical and computationally efficient path to a more expressive posterior that captures both instance-level and dimension-level dependencies within a unified framework.

---

> > ### Comment · Reviewer_Evn6 · 2025-08-04
> >
> > I thank the authors for their detailed and comprehensive rebuttal. The clarifications provided, along with the additional experimental results, have successfully addressed my concerns. As a result, I am happy to raise my score.

---

### Official Review · Reviewer_XNGw · 2025-07-02

**Clarity:** 3
**Significance:** 3
**Originality:** 3
**Rating:** 5
**Confidence:** 3

**Summary:**

Within the context of variational inference (VI) for latent-variable models, this paper proposes a variational structure that builds in dependence between k neighbouring latent indicators. In principle, doing so will allow improved dependence (e.g. temporal) modelling within the latents, and better modelling and predictive abilities on the observed data. The paper uses a Gaussian distribution for the variational approximation, but with some conditioning and reparameterising tricks, the ability to handle this computationally is managed, with reasonable scalability. The simulations show improved performance in a number of modelling tasks compared to existing methods with weaker dependence among latents.

**Questions:**

I was looking for some information about the computational overheads needed for this method (given that this is effectively the problem the paper is solving). Both for the presented simulations in Section 4 (comparative times for analysis of each method), but also an understanding (empirical and general) of how these overheads scale as k increases in a given computational task. Is k=3,5,10 the limit that this method can reasonably tackle, or is k=50,100,250 possible too?


Some trivial typos (no response required):

Fig 1 - should the last node be Z_N?
l.97 "connection structure [is] imposed" - delete "is"
l.102 "follows" -> "follow"
l.138 "for THE ELBO" insert "the"
l.146 "above TO BE valid." insert "to be"
l.298 and l.300 "state of THE GRU" insert "the"
l.319 "learn AN expressive" change "a" to "an"
l.327 Write L1, L2 as L_1, L_2, delete the extra space before the comma from "L_2 loss ,"
References lots of missing capitalisations. E.g. "lstm", "gaussian", "wasserstein", "bayesian", "Journal of machine learning research" etc.
Reference [57] and [58] look suspiciously similar. Is there a publication racket going on here?

**Ethical Concerns:**

["NO or VERY MINOR ethics concerns only"]

**Final Justification:**

The authors have cleared up some areas that were initially unclear, provided extra simulation to demonstrate method performance in more challenging scenarios, and done a proper cost analysis of computational overheads. Overall the method is a nice contribution, and (if the revisions are made), it should be well received in the community. I have changed my score to 5 to clearly indicate support to the AC.

**Limitations:**

Yes, but as discussed above, it would be good to have some further discussion on this, as it was only mentioned very briefly.

**Paper Formatting Concerns:**

None.

**Quality:**

3

**Strengths And Weaknesses:**

Strengths: The paper has seemingly found a way to build structure between latent variables, via a normal variational density, while managing to control the computational overheads for increasing dimension of dependence. This structure not only includes a k-Markov correlation structure, but this can be reworked to encompass a "tree-structured backbone" dependency structure.

The simulation experiments demonstrated that capturing such dependence is (of course) beneficial in a number of time series and clustering tasks.

Weaknesses: It seems that there may be some complexities involved in constructing more detailed correlation structures. I didn't get too much of a sense how hard or involved this was from the paper though.

It's also not clear what the computational overheads of the proposed method are, and how this scales with increasing k.

---

> ### Author Rebuttal · Authors · 2025-07-31
>
> We thank the reviewer for the comments and suggestions. Below we respond to each point raised by the reviewer.
>
> **Q1: How difficult or involved is it to construct more detailed correlation structures within the proposed framework?**
>
> A1: Constructing more detailed correlation structures within our framework is straightforward by design. We can increase detail in two main ways: (1) by increasing the approximation order k to capture higher-order correlations, (2) by generalizing the dependency backbone from a simple chain to a tree structure to model more flexible correlations among data instances.
>
> **Q2: What are the specific computational overheads of the proposed method? How do these overheads scale as the approximation order k increases?**
>
> A2: The specific computational overheads involve three parts: (i) the overhead of neural network evaluations for re-parameterizing correlation coefficients, (ii) the overhead of sampling from the variational posterior, and (iii) entropy calculation.
>
> In our method, to define a k-order posterior over $N$ latent variables, we need to estimate exactly $(N – 1)$ first-order, $(N – 2)$ second-order, and so on, up to $(N – k)$ k-order correlations, yielding a total of $(N-1) + (N-2) + \cdots + (N-k) = k(2N-k-1)/2$ correlation coefficients. Each coefficient is produced by a single forward pass through our re-parameterization network $f_{\phi}$, so this cost amount to the cost of evaluating $\mathcal{O}(kN)$ times of the neural network $f_{\phi}$.
>
> The sampling and entropy calculations involve operations like inversion and determinant computation on $k \times k$ sub-matrices, incurring a cost of $\mathcal{O}(k^3)$ FLOPs. Considering that $k$ is typically much smaller than $N$ and the complexity of evaluating neural networks, the cost of these operations is negligible compared to the cost of neural network evaluations.
>
> Therefore, the total cost of the variational inference approximately amounts to the cost of evaluating $\mathcal{O}(kN)$ times of the neural network $f_{\phi}$ per epoch, which scales linearly with the approximation order k.
>
> **Q3: What are the practical limits for k? Is k=10 the reasonable maximum, or are much larger values like k=50, 100, 250 also feasible?**
>
> A3: Yes, larger values of k beyond 10 are also feasible for our method, and we have conducted experiments with approximation orders up to k=50 and k=100, as shown in Tables 1-3 below.
>
> ---
>
> Table 1: Time series anomaly detection performance on SMAP dataset.
> | Methods  | Mean-field | k=1 | k=3 | k=10 | k=50 | k=100 |
> | - | - | - | - | - | - | - |
> | Runtime (s/epoch) | 1.00 | 2.44 | 8.51 | 27.60 | 142.58 | 294.53 |
> | F1     | 0.7774 | 0.8411 | 0.8552 | 0.8636 | 0.8711 | 0.8755 |
> | ELBO   | -109.2182 | -97.6057 | -95.2314 | -92.2948 | -90.4291 | -89.9577 |
>
> ---
>
> Table 2: Time series forecasting performance on ETTh1 dataset with horizon 24.
> | Methods  | Mean-field | k=1 | k=3 | k=10 | k=50 | k=100 |
> | - | - | - | - | - | - | - |
> | Runtime (s/epoch) | 4.80 | 11.45 | 36.94 | 126.44 | 675.75 | 1340.22 |
> | MSE | 0.739 | 0.664 | 0.543 | 0.363 | 0.348 | 0.333 |
> | MAE | 0.716 | 0.570 | 0.505 | 0.376 | 0.362 | 0.352 |
>
> ---
>
> Table 3: Constrained clustering performance on MNIST dataset.
> | Methods  | Mean-field | TreeVI | k=1 | k=3 | k=5 | k=10 | k=50 | k=100 |
> | - | - | - | - | - | - | - | - | - |
> | Runtime (s/epoch) | 0.25 | 0.52 | 0.59 | 1.84 | 3.09 | 6.22 | 29.53 | 60.59 |
> | ACC (\%)     | 96.50 | 97.43 | 97.55 | 98.12 | 98.39 | 98.52 | 98.62 | 98.69 |
> | NMI (\%)     | 91.37 | 93.14 | 93.44 | 93.80 | 94.17 | 94.55 | 94.63 | 94.85 |
> | ARI (\%)     | 92.54 | 93.73 | 93.89 | 94.89 | 95.24 | 95.65 | 95.85 | 96.09 |
>
> ---
>
> The results show that performance continues to improve beyond k=10, but with diminishing marginal returns. Since the runtime scales linearly with k, the significant increase in computational cost for these very high orders may not be justified by the smaller performance improvements. Therefore, we set k to moderate values (up to 10) in our main experiments, as it provides substantial performance gains over lower-order methods while maintaining computational efficiency. Ultimately, the optimal choice for k depends on the specific application and the available computational budget.

---

> > ### Comment · Reviewer_XNGw · 2025-08-04
> >
> > Thank you - for each of these I would be expecting that the “new information” is incorporated into the revised version (main text or supporting information) so that the next reader doesn’t have exactly the same questions. But the responses don’t indicate that this is the case, and seem to be just answering questions on this forum. Could the authors confirm that changes will actually be made to the paper?
> >
> > 1. Thank you. This should both be clearly stated, and perhaps the extra page in the revised version devoted to explaining the flexibility and capability of the tree-structure. I confess I found it difficult to imagine from what was written in Section 2.3 building a range of clear dependence structures based on the user’s design. It was just too compact. Just saying “yes it’s easy” here isn’t particularly helpful!
> >
> > 2. Computational overheads. Thank you for this (and for the empirical run times). I’d be tempted to keep the O(k^3) term around as you have k=100 in the paper (which could go higher), and for complex dependencies, it might not be too long before k^3 becomes noticeable. (Also, you can see from the run times that there is minor deviation away from O(kN) dominance for some datasets (SMAP, ETTh1; MNIST is spot on O(kN)). Presumably this discussion will be included in the revised paper?
> >
> > 3. k=100. Thank you. Presumably the larger k experiments will appear at least in the Supplementary Material? Stopping at k=10 in the paper isn’t particularly convincing without the extended results.

---

> ### Author Response · Authors · 2025-08-04
>
> Thank you for your constructive feedback. We confirm that we will incorporate the clarifications from our previous response into the revised version of our paper to improve clarity and completeness, considering that one additional page is allowed.
>
> First, we will expand our explanation of how the correlation structure is built upon a tree backbone. The initial step is to establish this backbone, for which there are several methods. A straightforward approach is to compute a similarity matrix between all data instances and then use a classic algorithm like a DFS to sample a tree. There are also some other approaches, such as the continuous optimization framework proposed by TreeVI, which directly learns the adjacency matrix $\mathbf{A}$ subject to an acyclicity constraint that guarantees the learned structure to represent a tree. In our paper, we adopt the simpler and more direct approach of using a similarity matrix. Once this tree backbone is established, our k-order structure is imposed upon it by defining correlations over the local (k+1)-vertex cliques on the tree.
>
> Additionally, as you suggested, we will add the detailed discussion on computational overheads to our main text and include the full experimental results for larger k values (up to k=100) in the supplementary material.

---

> > ### Comment · Reviewer_XNGw · 2025-08-06
> >
> > Thank you to the Authors for answering my questions, and committing to improve the paper in the ways discussed. After additionally reading the other reviews and responses, and to provide clarity to the AC, I am happy to raise my score to 5.

---

### Official Review · Reviewer_oEn6 · 2025-07-02

**Clarity:** 3
**Significance:** 3
**Originality:** 3
**Rating:** 5
**Confidence:** 3

**Summary:**

The paper focusses on approximate Bayesian inference in correlated latent variable models using variational inference. The paper proposes a Gaussian variational family with (chain or tree) structured  precision matrices.

The paper provide a product decomposition theorem (Theorem 2.1) for the density of a multivariate Gaussian, where the precision matrix exhibits k-order connectivity structure. The theorem also works "the other way", giving conditions for when the set of marginal form a valid joint density. The theorem enables efficient sampling from the joint distribution in a sequential manner via the set of marginals.

For efficiency, an amortized solution is proposed, where the required correlation coefficents of the variatonal distribution are reparametrized via a neural network. Specifically, the correlation matrices are re-parametrized through a combination of (constrained) first order and higher order correlation coefficient to ensure positive definiteness and proves that the resulting correlation matrix is valid (Theorem 2.2 and Corollary 2.3).

The paper concludes with an empirical evaluation of the proposed method in the context of time series anomaly detection, forecasting and constrained clustering. The results for anomaly detection shows that the proposed method outperforms the included baseline in terms F1 (better outlier detection) and ELBO (better representation of the exact posterior as measured by the KL divergence) and similar strong performance for the forecasting and clustering experiments

**Questions:**

I believe the paper could be stronger if  the authors could elaborate on the following points:

- The results generally show improved performance as a function of the order of the structured precision matrix. It is therefore natural to wonder whether the performance would be even better if the author went beyond 10? Did the authors experiment with approximation order beyond 10?

- Moreover, it would be interesting and informative to understand the effect of increasing the order of the approximation, all the way from mean-field to a fully flexible Gaussian (possible on a smaller dataset) and possibly also comparing to other variational families, e.g. low rank Gaussians and flows.

- It would also be interesting to investigate the trade in computational complexity when the order is increased from mean-field to a full Gaussian. What happens to the run time? When will it become more efficient to simply use a full Gaussian rather than the structured approximation.

-  There are no error bars/uncertainties associated with the metrics. How robust are the results?

- What is the experimental set-up for the baselines? Did the authors do an effort to make them appear as strong as possible or did they use them blindly?

- What is the likelihood of the VAE? A Gaussian? How many posterior samples were used during training and evaluation?

**Ethical Concerns:**

["NO or VERY MINOR ethics concerns only"]

**Final Justification:**

The detailed response from the authors addressed most of my questions/concerns, and therefore, I believe the paper should be accepted, and hence, my final score is 5.

**Limitations:**

yes

**Quality:**

3

**Strengths And Weaknesses:**

The paper is well-structured, well-written and easy to follow. The proposed method for bridging the gap between a mean-field posterior and a general Gaussian is interesting. There is a lot of prior work on structured precision matrices in general, but I believe that the proposed idea is novel in the context of variational inference (at least to the best of my knowledge) and will be of interest to the community.

Generally, the experimental design seems sound and shows a clear benefit relative to the included baselines. However, I am not that familiar with the set of included baselines and therefore, it is no clear to me whether they are strong baseline or easy to beat. Moreover, it is unclear how the baseline methods are configured, to what degree they are tuned etc.

Moreover, some of the implementation details are unclear. For example, what is the likelihood/loss in the VAE? It is a standard Gaussian? How many posterior samples is used for 1) training and 2) evaluation? Finally, the authors do not report uncertainty for any of the evaluation metrics, i.e. no error bars etc.

---

> ### Author Rebuttal · Authors · 2025-07-31
>
> We thank the reviewer for the comments and suggestions. Below we respond to each point raised by the reviewer.
>
> **Q1: Are the chosen baselines strong or merely easy to beat? How were they configured and tuned during experimentation?**
>
> A1: Our chosen baselines represent strong and state-of-the-art methods in their respective domains, selected to ensure a fair evaluation of our model. For the time series modeling tasks, we compared our model against both leading generative methods and prominent forecasting models. For constrained clustering, we benchmarked against state-of-the-art algorithms.
>
> To ensure a fair comparison, we configured and tuned all methods to achieve their best possible performance. Hyperparameters of baseline methods are set consistent with our method as far as possible.
>
> **Q2: What is the specific likelihood or loss function used in the Variational Autoencoder (VAE) component? And how many posterior samples were utilized during both the training and evaluation phases of the model?**
>
> A2: The specific likelihood function $p( \mathbf{X} |\mathbf{Z})$ used in our VAE is task-dependent. For the time series tasks, the likelihood is a Gaussian distribution with parameters produced by a GRU-based decoder. For constrained clustering, the likelihood is a normal distribution conditioned on a cluster-dependent latent variable. Across all applications, the loss function is the negative of the ELBO. For all experiments, we utilized a single sample from the posterior distribution during both training and evaluation.
>
> **Q3: Why are there no error bars or uncertainties reported for the evaluation metrics, and how robust are the presented results without them?**
>
> A3: We would like to clarify that we did report error bars in Table 3 across 10 independent runs. For Tables 1 and 2, we omitted the error bars in the main paper due to space constraints, and we will include them in the appendix in our revised manuscript. To address your concern directly, we present the F1-Scores and ELBOs for the time series anomaly detection task with their corresponding standard deviations from 10 runs below:
>
> |      |      |        k=1       |        k=3       |       k=10       |
> | - | - | - | - | - |
> | SMAP |  F1  |   0.8411±0.0078  |   0.8552±0.0070  |   0.8636±0.0065  |
> |      | ELBO |  -97.6057±4.4278 |  -95.2314±3.4668 |  -92.2948±3.835  |
> |  MSL |  F1  |   0.8883±0.0070  |   0.8940±0.0067  |   0.9145±0.0064  |
> |      | ELBO | -165.5004±3.9339 | -157.2134±3.7234 | -134.0815±3.7679 |
> |  SMD |  F1  |   0.8901±0.0084  |   0.9153±0.0081  |   0.9284±0.0087  |
> |      | ELBO |  -69.5278±1.3356 |  -67.4001±1.5207 |  -65.0345±1.2238 |
>
> **Q4: Were experiments conducted with approximation orders exceeding 10? If so, did performance continue to improve at higher orders?**
>
> A4: Yes, we have conducted additional experiments with orders k exceeding 10, specifically for k=50 and k=100, to explore the performance limits of our model, as shown in Tables 1-3 below.
>
> ---
>
> Table 1: Time series anomaly detection performance on SMAP dataset.
> | Methods  | Mean-field | k=1 | k=3 | k=10 | k=50 | k=100 |
> | - | - | - | - | - | - | - |
> | Runtime (s/epoch) | 1.00 | 2.44 | 8.51 | 27.60 | 142.58 | 294.53 |
> | F1     | 0.7774 | 0.8411 | 0.8552 | 0.8636 | 0.8711 | 0.8755 |
> | ELBO   | -109.2182 | -97.6057 | -95.2314 | -92.2948 | -90.4291 | -89.9577 |
>
> ---
>
> Table 2: Time series forecasting performance on ETTh1 dataset with horizon 24.
> | Methods  | Mean-field | k=1 | k=3 | k=10 | k=50 | k=100 |
> | - | - | - | - | - | - | - |
> | Runtime (s/epoch) | 4.80 | 11.45 | 36.94 | 126.44 | 675.75 | 1340.22 |
> | MSE | 0.739 | 0.664 | 0.543 | 0.363 | 0.348 | 0.333 |
> | MAE | 0.716 | 0.570 | 0.505 | 0.376 | 0.362 | 0.352 |
>
> ---
>
> Table 3: Constrained clustering performance on MNIST dataset.
> | Methods  | Mean-field | TreeVI | k=1 | k=3 | k=5 | k=10 | k=50 | k=100 |
> | - | - | - | - | - | - | - | - | - |
> | Runtime (s/epoch) | 0.25 | 0.52 | 0.59 | 1.84 | 3.09 | 6.22 | 29.53 | 60.59 |
> | ACC (\%)     | 96.50 | 97.43 | 97.55 | 98.12 | 98.39 | 98.52 | 98.62 | 98.69 |
> | NMI (\%)     | 91.37 | 93.14 | 93.44 | 93.80 | 94.17 | 94.55 | 94.63 | 94.85 |
> | ARI (\%)     | 92.54 | 93.73 | 93.89 | 94.89 | 95.24 | 95.65 | 95.85 | 96.09 |
>
> ---
>
> The results show that performance does continue to improve as the order k increases beyond 10, which further validates our central claim that capturing higher-order dependencies is beneficial.
>
> However, we observed that the marginal performance gains diminish as k becomes excessively large, and concurrently the computational runtime increases linearly with the order k. We focused on k up to 10 in the main paper because this range demonstrates the advantages of our approach while maintaining a practical and efficient balance between performance and computational tractability.
>
> **Q5: What is the effect of systematically increasing the approximation order from a mean-field approach all the way to a fully flexible Gaussian (potentially on a smaller dataset)? How does the proposed method compare to other variational families, such as low-rank Gaussians and normalizing flows?**
>
> A5: Increasing the approximation order k consistently improves model performance by capturing higher-order correlations, as seen from Tables 1-3. To capture all correlations, k would need to equal $N-1$, making our model equivalent to a fully flexible Gaussian. However, the computational cost scales linearly with k, with the runtime being approximately k times that of a mean-field method. This makes a fully flexible Gaussian computationally intractable for instance-level correlation modeling, even on smaller datasets.
>
> Since other variational families like low-rank Gaussians and normalizing flows are not designed for instance-level modeling, we apply our method to model dimension-level correlations within the latent space of a VAE to provide a fair comparison. We configured a VAE with 30 latent dimensions on MNIST and report the ELBO on the test set below.
>
> | Method              | Test ELBO |
> |---------------------|:---------:|
> | Mean-Field Gaussian |   -84.62  |
> | Low-Rank Gaussian (rank=10)  |   -83.15  |
> | Normalizing Flow (10 steps of planar flow)   |   -82.51  |
> | Ours (k=3)          |   -83.54  |
> | Ours (k=5)          |   -82.91  |
> | Ours (k=10)         |   -82.38  |
> | Ours (k=30, Full-Rank Gaussian) |   -81.73  |
>
> The results show that our method bridges the gap between mean-field approximations and fully expressive models. Even with a small order (k=3), our method outperforms the mean-field approach and becomes competitive with the low-rank Gaussian. As k increases to 10, our method surpasses both the low-rank Gaussian and the normalizing flow in this setup, and approaches the performance of the full-rank Gaussian, demonstrating that it offers a competitive and highly flexible alternative to other variational families.
>
> **Q6: What is the trade-off in computational complexity and runtime when the approximation order is increased from mean-field to a full Gaussian? At what point does it become more computationally efficient to simply use a full Gaussian rather than the structured approximation?**
>
> A6: The choice of the approximation order k is a direct trade-off between model expressiveness and computational cost. As k increases, our model captures higher-order correlations, leading to consistently better performance, as shown in Tables 1-3 above. However, the computational cost also scales linearly with k, with the runtime being approximately k times that of a mean-field model. Because of this trade-off between diminishing performance gains and linearly increasing cost, we selected a moderate k (up to 10) in our main experiments to strike a balance between performance and efficiency.
>
> When k is set to $N-1$, where $N$ is the number of instances, our method becomes equivalent to a full Gaussian. However, a full Gaussian would require evaluating $\mathcal{O}(N^2)$ times of neural networks $f_\phi(\cdot)$ per epoch, which is computationally intractable for instance-level correlation modeling, especially considering that the number of instances can be very large in practice.

---

> > ### Comment · Reviewer_oEn6 · 2025-08-06
> > **Thank you for the response**
> >
> > Thank you for the detailed response. It addresses most of my questions/concerns, and therefore, I'll keep my score of 5 (accept).

---

### Official Review · Reviewer_yDoz · 2025-07-07

**Clarity:** 2
**Significance:** 3
**Originality:** 2
**Rating:** 5
**Confidence:** 3

**Summary:**

The authors propose a more accurate variational inference approximation that takes instance level correlation into account. They do this by approximating the N by N covariance matrix with a K by K matrix that takes into account the correlation between $i$th instance upto the $i+k$th instance. The correlation coefficients of this K by K matrix are parametrised through a neural network. The authors then introduce a method to ensure that the resultant covariance matrix is PD as well as an extension to a tree based correlation structure. The authors show improved performance in time series and constrained clustering tasks.

**Questions:**

- What is the intuition behind the K level correlation modelling? Is using a value of 1 effectively a Markov assumption in the posterior?
- At what level of K should we recover the same correlation structure as the baselines? In Table 1 I would have expected the 1 order method to perform as well as an LSTM-VAE that models the same correlation (apart form the fact that the authors use a GRU).
- Why is it necessary to directly parametrise the correlation coefficients $R$? If terms of the covariance matrix are parametrised won't this allow for parametrising the cholesky of the covariance matrix which will ensure that the covariance is PD by construction?
- What kind of neural network is $f_{\phi}$? Are the different $x_i$ concatenated across the feature dimension or is attention applied across the $K$ dimensions?
- Is there any relation to the inducing points framework used in sparse GPs? The problem being solved seems similar, here an N by N matrix is too complex to invert, so it is approximated by a K by K matrix with K < N.
- What is the cost of the sampling procedure laid out in L121?

**Ethical Concerns:**

["NO or VERY MINOR ethics concerns only"]

**Final Justification:**

The authors have addressed my concerns and I don't have any other major concerns.

**Limitations:**

Yes

**Quality:**

3

**Strengths And Weaknesses:**

**Strengths:**
- The method is interesting and sound.
- The experiment are fairly wide ranging and over multiple baselines.

**Weaknesses:**
- The paper could be a lot clearer:
     - What is the exact cost of inference with P?
     - The intuition behind modelling only K correlations (apart from better computational cost) is not clear. What are you losing here? There should also be a discussion on the intuition for choosing this parameter.
    - The baselines are not sufficiently described. Which of these methods actually model instance level correlation? If they do, what is the difference between the proposed method?

---

> ### Author Rebuttal · Authors · 2025-07-31
>
> We thank the reviewer for the comments and suggestions. Below we respond to each point raised by the reviewer.
>
> **Q1: What is the exact cost of inference with $P$?**
>
> A1: The exact cost of inference with our proposed k-order precision matrix $P$ involves three parts: (i) the cost of neural network evaluations for re-parameterizing correlation coefficients, (ii) the cost of sampling from the variational posterior, and (iii) entropy calculation.
>
> In our method, to define a posterior with k-order correlation over $N$ latent variables, we need to specify exactly $(N–1)$ first-order, $(N–2)$ second-order, and so on, up to $(N–k)$ k-order correlations, yielding a total of $(N-1)+(N-2)+\cdots+(N-k)=k(2N-k-1)/2$ correlation coefficients. In our method, each coefficient is parameterized by the output of a re-parameterization network $f_{\phi}(\cdot)$. Therefore, to re-parameterize these coefficients, we need to run the network $f_{\phi}(\cdot)$ for $\mathcal{O}(kN)$ times.
>
> The sampling and entropy calculations involve operations like inversion and determinant computation on $k \times k$ sub-matrices, incurring a cost of $\mathcal{O}(k^3)$ FLOPs. Considering that k is typically much smaller than $N$ and the complexity of evaluating neural networks, the cost of these operations is negligible compared to the cost of neural network evaluations.
>
> Therefore, the total cost approximately amounts to the cost of evaluating $\mathcal{O}(kN)$ times of the neural network $f_{\phi}$ per epoch, which is approximately k times of the cost of mean-field amortized VI methods.
>
> **Q2: What is the intuition behind k-level correlation modeling? How should $k$ be chosen? What are you losing here? Does k=1 imply a Markov assumption in the posterior?**
>
> A2: The choice of k is a trade-off between model expressiveness and computational cost. Generally, as k increases, the model's performance consistently improves, as demonstrated in our experimental results. However, as seen from Table 1 below, the performance gains diminish as the order k (e.g. 50, 100) goes higher. Meanwhile, the computational cost always scales linearly with k according to our analysis in answer A1. To balance the gains and cost, we set k to moderate values (up to 10) in our main experiments.
>
> By setting k to a moderate value (e.g. 10), we can only model correlation up to 10-th order, losing the ability to model higher-order correlations. But as observed from Table 1, the gains become increasely weak as the order goes higher. Therefore, setting k to a moderate value is a reasonable choice.
>
> Yes, when k=1, our variational posterior is reduced to the first-order Markov random field distribution.
>
> ---
> Table 1: Time series anomaly detection on SMAP dataset.
> |Methods|Mean-field|k=1|k=3|k=10|k=50|k=100|
> |-|-|-|-|-|-|-|
> |Runtime(s/epoch)|1.00|2.44|8.51|27.60|142.58|294.53|
> |F1|0.7774|0.8411|0.8552|0.8636|0.8711|0.8755|
> |ELBO|-109.2182|-97.6057|-95.2314|-92.2948|-90.4291|-89.9577|
>
> **Q3: Which baselines actually model instance-level correlation, how do they differ from the proposed method?**
>
> A3: Due to the difficulties of modeling instance-level correlation, existing works mostly only consider the modeling of correlation within one instance. In our baselines, only OmniAnomaly and TreeVI explicitly model instance-level correlations in their posterior distributions.
>
> OmniAnomaly imposes a first-order Markov assumption among latent variables and employs an MLP to parameterize the one-step conditional Gaussian, where $(\mu_{z_t},\sigma_{z_t})=\text{MLP}([z_{t-1},e_t])$. This design makes it essentially a recurrent model, just like an RNN learning a transition function for its hidden state. As a result, it is inherently limited to modeling only first-order correlation. More importantly, OmniAnomaly can only be applied to model chain-structured correlation due to its inherent RNN-like structure, making it unable to model correlation with more complicated structure like trees. In contrast, our method not only can model higher-order correlation, but also can be applied to model correlations with tree backbones.
>
> The key difference between TreeVI and our method is that TreeVI is inherently limited to modeling only first-order correlations, while our method can capture higher-order correlations. This limitation of TreeVI arises from its reliance on an acyclic tree structure to construct its correlation matrix. While this is sufficient for simple pairwise relationships, attempting to model higher-order correlations inherently introduces loops into the underlying correlation structure. The construction of TreeVI depends on the acyclic property of its backbone and is no longer valid when these loops exist. Moreover, simply modeling higher-order correlation coefficients within the framework of TreeVI does not guarantee the correlation matrix to be positive definite. So even though TreeVI can capture instance-level correlations, it cannot be easily extended to model higher-order correlations.
>
> **Q4: At what level of k should the proposed method recover the same correlation structure and performance as baselines? In Table 1, I would have expected the 1 order method to perform as well as LSTM-VAE.**
>
> A4: At k=1, our method is reduced to TreeVI, with both methods modeling first-order instance-level correlation in the same way. But for other baselines, although some of them also attempt to model first-order instance-level correlation, they are fundamentally different from us in terms of correlation modelling methods and the ability of generalizing to more complicated correlation structures.
>
> As for the performance gap between LSTM-VAE and our method, the reason is explained below. Although LSTM-VAE also attempts to capture instance-level correlation, it is achieved via the deterministic dependency among the hidden states in RNN. Given the hidden states' values, the latent variables $z_i$ of instances at different time steps are independent. By contrast, our method not only captures the deterministic dependency, but also models the correlation among the stochastic variations of latent variables $z_i$ corresponding to different instances. That is,  given the hidden states' values, in our method, the latent variables $z_i$ are still correlated. Thus, our method is able to capture more accurate and complicated correlation among data. That explains why our method  performs better than LSTM-VAE even under the case of k=1.
>
> **Q5: Why is it necessary to directly parameterize the correlation coefficients $R$? Why not parameterize the Cholesky decomposition of the covariance matrix to ensure positive definiteness by construction?**
>
> A5: We first want to emphasize that this work aims to capture instance-level correlation, under which the dimension of correlation matrix is $N\times N$, with $N$ denoting the number of instances. Parameterizing the Cholesky factor $\mathbf{L}$ of a correlation matrix $\mathbf{R} = \mathbf{L} \mathbf{L}^\top$ does ensure its positive definiteness. However, we have to re-parameterize as many as $N(N+1)/2$ elements, i.e., all elements from the lower-triangular positions of $\mathbf{L}$ need to be re-parameterized. That means we need to evaluate the neural network $f_\phi(\cdot,\cdot)$ by $\mathcal{O}(N^2)$ times for every epoch, which is computationally unacceptable, especially considering that $N$ could be very large in practice.
>
> To reduce the cost, a common practice is to restrict $\mathbf{L}$ to be low-rank, which reduces the number of parameters to be re-parameterized from $\mathcal{O}(N^2)$ to $\mathcal{O}(kN)$ for a low rank $k$. However, each element in $\mathbf{L}$ does not explicitly correspond to the correlation between any two specific data instances. Therefore, to parameterize each element in $\mathbf{L}$, we have to feed all $N$ data instances into the neural network $f_\phi$. Obviously, this will result in the network to be huge, and thereby incurring unacceptable computation and storage burden.
>
> In contrast, although our method also need to re-parameterize $\mathcal{O}(kN)$ parameters, in our method, the parameter is the correlation coefficient. For each correlation coefficient, such as $\gamma_{i,i+t}^c$, it directly represents the correlation between a specific pair of instances $(x_i,x_{i+t})$. Therefore, we only need to feed the pair of instances into the neural network $f_\phi$, making the network to be small.
>
> **Q6: What kind of neural network is $f_{\phi}$?**
>
> A6: The function $f_{\phi}$ is implemented as a standard MLP. Inputs $x_i$ and $x_{i+t}$ are concatenated across the feature dimension to form a single input, which is then processed by the MLP with a tanh activation function to constrain the output to the range of (-1,1).
>
> **Q7: Any relation to inducing points used in sparse GPs, given that both aim to approximate large matrices?**
>
> A7: While both our method and the inducing points method aim for scalability, the inducing points framework cannot handle instance-level correlation modeling effectively. The inducing points method works by a low-rank approximation based on a small set of $M$ representative points. Since these points must summarize the properties of the entire dataset, the network used to learn their optimal locations needs to see all $N$ data instances to derive this global summary. This results in a network with an $\mathcal{O}(N)$ input size, which is computationally intractable for large datasets. Therefore, low-rank methods like inducing points are not usable in our setting. In contrast, in our method, the re-parameterization neural network used to compute each correlation coefficient only requires a pair of instances as input.
>
> **Q8: Computational cost of the sampling procedure laid out in line 121?**
>
> A8: The cost is composed of two parts: $\mathcal{O}(kN)$ times of neural network evaluations for re-parameterizing correlation coefficients, and $\mathcal{O}(k^3)$ FLOPs for matrix inversion. For more details, please refer to A1.

---

> > ### Comment · Reviewer_yDoz · 2025-08-06
> > **Response to authors**
> >
> > I thank the authors for their response. They have addressed my concerns.

---

### Decision · Program_Chairs · 2025-09-17

**Decision:**

Accept (poster)

**Comment:**

We thank the authors for their submission.

The authors propose a structured variational approximation that enables structured covariance between latent variables while remaining tractable, and providing a direct tradeoff between fidelity and computational efficiency.  Reviewers voiced concerns over potential runtime issues, which the authors addressed throughout the author-reviewer discussion.  In a somewhat crowded area, the proposed method strikes a novel balance between efficiency and expressivity.  In particular, this work enables the capturing of higher order correlations, which is empirically shown to improve performance.